

# Introducing a new floodplain scheme in ORCHIDEE (version 7885): validation and evaluation over the Pantanal wetlands

Anthony Schrapffer[1,2,3,4], Jan Polcher[5], Anna Sörensson[1,2,3], and Lluís Fita[1,2,3]

[1]Universidad de Buenos Aires, Facultad de Ciencias Exactas y Naturales. Buenos Aires, Argentina.
[2]CONICET – Universidad de Buenos Aires. Centro de Investigaciones del Mar y la Atmósfera (CIMA). Buenos Aires, Argentina.
[3]CNRS – IRD – CONICET – UBA. Instituto Franco-Argentino para el Estudio del Clima y sus Impactos (UMI 3351 IFAECI). Buenos Aires, Argentina.
[4]EthiFinance, 11 Avenue Delcassé 75008 Paris.
[5]Laboratoire de Météorologie Dynamique (LMD), IPSL, CNRS, École Polytechnique, Palaisseau, France.

**Correspondence:** Anthony Schrapffer (anthony.schrapffer@gmail.com)

**Abstract.** Adapting and improving the hydrological processes in Land Surface Models is crucial given the increase of the resolution of the Climate Models to correctly represent the hydrological cycle. The present paper introduces a floodplains scheme adapted to the higher resolution river routing of the ORCHIDEE Land Surface Model. The scheme is based on a sub-tile parameterization of the hydrological units, Hydrological Transfer Unit concept (HTUs), based on high resolution

hydrologically-coherent Digital Elevation Models which can be used for all types of resolutions and projections. The floodplain scheme was developed and evaluated for different atmospheric forcings and resolutions (0.5° and 25km) over one of the world's largest floodplains: the Pantanal, located in Central South America.

The floodplains scheme is validated based on the river discharge at the outflow of the Pantanal which represents the hydrological cycle over the basin, the temporal evolution of the water mass over the region assessed by the anomaly of Total Water

Storage in Gravity Recovery And Climate Experiment (GRACE) and the temporal evaluation of the flooded areas compared to the Global Inundation Extent from Multi-Satellites dataset (GIEMS-2). The hydrological cycle is satisfactorily simulated, however, the base flow may be underestimated. The temporal evolution flooded area is coherent with the observations although the size of the is underestimated in comparison to GIEMS-2.

The presence of floodplains increases the soil moisture up to 50% and decreases average temperature with 3°C and with

6°C during the dry season. The higher soil moisture increases the vegetation density and, along with the presence of open water surfaces due to the floodplains, it affects the surface energy budget by increasing the latent flux at the expense of the sensible flux. This is linked to the increase of the evapotranspiration related to the increased water availability. The effect of the floodplains scheme on the land surface conditions highlights that coupled simulations using the floodplains scheme may influence local and regional precipitation and regional circulation.





## 1 Introduction

Floodplains are areas adjacent to rivers which are seasonally flooded due to the overflow of rivers. They are particular places of interaction between the river network, the land surface conditions and the atmosphere because they can evaporate the water from the precipitation over the upstream area, i.e. non-local water. For this reason, the floodplains scheme of the Organising Carbon and Hydrology In Dynamic Ecosystems (ORCHIDEE, https://orchidee.ipsl.fr/ Krinner et al., 2005) model has been adapted

to the new high resolution river routing with particular objectives to: (1) better understand the land-atmosphere interactions over the floodplains and (2) further integrate it in high resolution coupled simulations using the Regional Earth System Model (RESM) of the Institut Pierre Simon Laplace regional climate model. The high resolution river routing is described more in detail in Polcher et al. (2022).

Climate modeling is heading toward higher resolution models, whether it concerns Global Climate Models (GCM) or Re-

gional Climate Models (RCM) because it allows representing the dynamic of the atmosphere with more details such as, for example, with the phase changes (concerning clouds and surface) which are critical for the water cycle and with the explicit representation of convection (Prein et al., 2015; Lucas-Picher et al., 2021). The land hydrological processes are important because they can strongly impact on the land-atmosphere feedbacks (Seneviratne and Stöckli, 2008; Dirmeyer, 2011; Seneviratne et al., 2010). The resolution of Land Surface Models has therefore also been increasing over the past decades and, in some

cases, their respective river routing schemes have been adapted to better fit these configurations (Guinaldo et al., 2021; Munier and Decharme, 2021; Chaney et al., 2021). Higher resolution models improve the land-atmosphere interactions by allowing the representation of smaller scale processes in LSMs (Barlage et al., 2021; Stephens et al., 2023). Small scale features will also have an increased importance at higher resolution (Stephens et al., 2023). Therefore, LSMs will be required to integrate more hydrological processes and to reconsider the processes already available to adapt them at these smaller scales. Most of

these processes are related to lateral water movements in relation with the river network such as floodplains, dams, lakes or the irrigation. As resolution increases they cannot be treated as sub-grid anymore. This effort is also valuable to better represent other climate related issues such as food and energy production as well as freshwater supply which are strategic issues for human adaptation to climate change (Karabulut et al., 2016; Bazilian et al., 2011; Howells et al., 2013). Beyond that, some of the hydrological features, such as the floodplains, are rich ecosystems whose natural equilibrium is fragile (Junk et al., 2006;

Bergier, 2013) and that can suffer from climate change (Bergier, 2013). Their representation in climate models is also crucial to evaluate how these regions will respond to climate change and if there is a risk of a tipping point at which these ecosystems would be permanently transformed (Thielen et al., 2020; Bergier, 2013).

Schrapffer et al. (2020) shows the importance of the evaporation of the non-local water surface in the *Pantanal*, a South American tropical floodplains. This process becomes more important at higher resolution as the horizontal gradients of the

surface conditions may play an important role at these resolutions and needs to have an adapted modeling to have an adequate spatial representation of the flooded areas. Moreover, in floodplains located in transition climate zone between wet tropical climate and semi-arid region such as the Pantanal, the extra-evaporation over the wetlands can generate strong horizontal gradients of land-atmosphere fluxes and affects both the local circulation and the regional precipitation patterns (Taylor, 2010;



Taylor et al., 2018; Adler et al., 2011). Therefore the representation of these features has an importance for climate models as
they (1) improve the realism of the local surface conditions and (2) influence the representation of the precipitation.

The representation of the river network and its relative processes in LSMs can be performed through (1) an hydrological model forced by the output of a Land Surface Model or (2) the integration of a river routing scheme within the LSM. In the first case, the hydrological models forced by the output of a LSM (CaMaFloods, Yamazaki et al., 2011; MGB-IPH, Collischonn et al., 2010; Paiva et al., 2011; Pontes et al., 2017; HyMAP, Getirana et al., 2020 or LISFLOOD-FP, Makungu and Hughes,
2021) generally use hydrologically coherent units (cf. vector-based representation in Yamazaki et al., 2013) and are, therefore, not constrained by an atmospheric grid. In the second case, most of the river routing scheme in LSMs are using a grid-based representation of the river network on a regular grid at a fixed resolution such as such as the ISBA-CTRIP 1/12° resolution routing used in Surfex (Guinaldo et al., 2021) and, therefore, are not flexible to the different projections / resolutions used in the coupled models and make interactions with the atmosphere more complex because they interpolate the output of the LSM
to the grid of the routing.

The forcing of an hydrological model from the output of a Land Surface Model based on a different grid can be performed through the interpolation of the runoff / drainage fluxes from the LSM to feed the Hydrological Model. In this case, the LSM doesn't necessarily receive the feedbacks from the hydrological model processes (cf. one-way coupling concept, Getirana et al., 2021) although, sometimes, it may receive some information such as the flooded area. In order to perform a two-way coupling,
there are 2 different solutions: (1) either the LSM and the hydrological model use the same grid or (2) water volume and open water surfaces are interpolated to simulate the feedbacks between the hydrological model and the LSM.

Originally, the river routing schemes integrated in most of the LSMs used a grid-based description of the river network. However, recent developments trend is toward a higher resolution description of the river network such as with the Hydrological Transfer Units concept in the ORCHIDEE model (HTUs; Polcher et al., 2022; Nguyen-Quang et al., 2018) or the Hydrologic
Response Units in the HydroBLOCKS model (HRU; Chaney et al., 2020). This description can be adapted to different atmospheric grids to facilitate the feedback between the LSM and the river routing scheme. In this case, the hydrological units are sub-tile units constructed from high resolution hydrological data (HDEM) such as MERIT-Hydro (Yamazaki et al., 2019) or HydroSHEDS (Lehner et al., 2008). The description of the river network is able to have hydrologically coherent units and to respect the atmospheric grid structure. This type of routing is referred to as a hybrid-based description of the river network
(Yamazaki et al., 2013). In ORCHIDEE, the HTU concept described in Nguyen-Quang et al. (2018) has been further improved and the HTUs can now be constructed with a flexible river routing pre-processor (Polcher et al., 2022).

The representation of the large scale floodplains in LSMs has been previously developed at 0.5° in the ORCHIDEE (Schrapffer et al., 2020; Lauerwald et al., 2017; Guimberteau et al., 2012; D'Orgeval, 2006), JULES (Dadson et al., 2010) and ISBA-CTRIP (Decharme et al., 2019) models. The relatively coarse resolution allowed these models to represent the floodplains with
a relatively simple parameterization as the floods can be handled locally within each hydrological unit which were at a 0.5° resolution. At higher resolution, the correct representation of the floodplains requires interactions and transfer of water between the different hydrological units and atmospheric grids to correctly simulate the lateral expansion of the floodplains (Getirana et al., 2021; Decharme et al., 2019).



More complex hydrological models such as CaMa-Floods, MGB-IPH, HyMAP and LISFLOOD-FP have a more precise
representation of the floodplains, in particular, this is due to their vector-based hydrological units, higher resolution and differ-
ent hydrological dynamic. They represent more precisely the flooded area within the hydrological units because they calculate
it from HDEM information by calculating the height of the river and the flooded area using the floodplains vertical profile
along the river based on descriptive variables such as the Height Above the Nearest Drainage variable (HAND, Nobre et al.,
2011). Apart from the previous difficulty to couple this type of model with a LSM, there are other difficulties such as (1)
the uncertainty of the orography in the HDEM over lowland areas such as the floodplains due to imprecision and vegetation
(Yamazaki et al., 2017), (2) the presence of divergent flows that are not integrated in the HDEM (Yamazaki et al., 2019).

Although the coupling between LSM and this type of hydrological model can improve the representation of the discharge
and of the flooded area, an efficient two-way coupling requires the use of the hydrological model on the same grid as the LSM
such as it is done in Marthews et al. (2021) and, therefore, limits the performance of the hydrological model. Moreover, the
interaction between both models is limited to some variables which complicates the possibility to integrate complex interactions
between the hydrological features and the soil hydrology and, therefore, can limit the process understanding.

The use of the high resolution routing scheme in ORCHIDEE based on the HTU concept has motivated the development of
an adapted floodplains scheme. The 0.5° resolution floodplains scheme developed by D'Orgeval (2006) has been reconsidered
to be adapted to higher resolutions and to different types of grid through the use of ORCHIDEE pre-processors RoutingPP
(Polcher et al., 2022) which generates the HTU graphs on the atmospheric grid. In this particular case, the higher resolution
of the hydrological units will exacerbate the difficulty to simulate the correct extent of floods due to the necessity to include
more complex water fluxes between the hydrological units. This is related to the fact that, in modeling, floods are usually well
estimated over the main river but underestimated over the adjacent areas (Decharme et al., 2019). The HTU representation is
useful to overcome this difficulty as (1) it gives the opportunity to define floodplains with more details and (2) the increased
information on river network connectivity allows to model the flooding of the area of floodplains which are adjacent to the
main rivers. Nevertheless, the floodplains scheme needs to be adjusted to the HTUs description by changing its dynamic and the
volume / flooded area relationship. The scheme developed in the present paper is complementary to other sources of information
to study large floodplains hydrology and surface conditions such as ground-based observations, satellite observations (Alsdorf
et al., 2010; Lee et al., 2011) and, in particular, satellite algorithms which have difficulty to estimate evapotranspiration due to
the presence of open-water surfaces (Penatti et al., 2015). This is why, apart from improving the surface conditions in LSM, the
development of a floodplains scheme at high resolution may also help to better understand the hydrological processes related
to the floodplains.

This article contains the description of a high resolution floodplains scheme for the ORCHIDEE Land Surface Model devel-
oped by the Institut Pierre Simon Laplace (IPSL), its validation and the analysis of its impact over the land surface variables
over the Pantanal floodplains. Section 2 describes the floodplains scheme as implemented in the river routing scheme of OR-
CHIDEE and the different equations ruling the exchange of water. The validation methodology and the observational data-sets
used are discussed in section 3. Section 4 and 5 then present the validation of the scheme. First it is performed on the variables
directly linked with the river routing scheme (discharge, flooded area, volume of water in the routing reservoirs). Secondly




the impact of the floodplains scheme on the land surface states and the land-atmosphere fluxes are evaluated. The assessment

and analysis of the impact of the floodplains scheme is performed based on simulations at different resolutions using a 0.5°

atmospheric forcing and a 20-km atmospheric forcing. The final section presents the discussion and conclusion.

## 2 Floodplains scheme description

The HTUs can be represented as a forest of directional rooted tree graphs (Foulds, 1992). Each tree has a root which is either

located at the coast (the river mouth) or in a lake when it is an endorheic basin. There cannot exist any loop in the river graph.

The graphs in the routing scheme are said to be convergent because each HTU only flows into a single HTU and is acyclic as

water cannot return to the original HTU.

Each HTU is fully contained in one atmospheric cell of the grid. The cells of ORCHIDEE can contain more than one HTU

and can be crossed by more than one river graph. The atmospheric grid of a HTU $i$ of the river graph is noted $\widehat{i}$. The surface of

an HTU $i$ is noted $S_i$ and the surface of $\widehat{i}$ is $S_{B,\widehat{i}} = \sum_{i \in \widehat{i}} S_i$.

The relations between the HTUs within the river graph are represented by an integer index. The natural flow direction of

the river is used to order the indices of the water stores on the graph where the index increases as the HTU are closer to the

river outflow. We note $\{i - 1\}$ the ensemble of all the upstream HTU of the HTU $i$. $i + 1$ is the unique downstream HTU of

the HTU $i$. The flux of water between the HTUs are placed on the edges of the graph and are indexed with half indices. Each

HTU is linked to an ensemble of upstream HTUs but only to one downstream. For example, the outflow of HTU $i$ is noted and

is part of the ensemble of inflow of the HTU $i + 1$: $i + 1/2 \in \{(i + 1) - 1/2\}$. The water flowing into the HTU $i$ is given by the

ensembles of fluxes on edges $\{i - 1/2\}$.

ORCHIDEE simulates the volume of water in the floodplains in each HTU $i$ (noted $V_{fp,i}$). This volume is then converted

into a flooded fraction $f_i$ based on the known potential flooded fraction for this HTU $f_{max,i}$. $f_{max,i}$ is obtained from the

Global Lake and Wetland dataset (GLWD; WWF, 2004), see subsection 2.5 for more details. The potential flooded surface for

an HTU $i$ is $S_{fmax,i} = S_i * f_{max,i}$. We consider that an HTU $i$ is a floodplains if $S_{fmax,i} > 0$. The actual flooded fraction and

the flooded surface are noted respectively $f_i \in [0, f_{max,i}]$ and $S_{f,i} \in [0, S_{fmax,i}]$. A more detailed description will be found in

subsection 2.4.

### 2.1 Floodplains fluxes

This subsection focuses on the definition of the different water fluxes between the floodplains and the atmosphere/soil. These

fluxes calculated for each HTU are (1) the precipitation over the flooded area ($P_{f,i}$), (2) the evaporation of the flooded area

($E_{f,i}$) and (3) the infiltration of the water in the floodplains into the soil moisture ($I_{f,i}$). These different fluxes are described

below.

The precipitation over the flooded area goes directly to the floodplains reservoir. Considering an HTU $i \in \widehat{i}$, the precipitation

going directly to the floodplains reservoir of this HTU ($P_{f,i}$) is described in equation 1.



$$P_{f,i} = P_{\widehat{i}}.S_{f,i}/S_{B,\widehat{i}} \tag{1}$$

with $P_{\widehat{i}}$ the precipitation over the grid grid cell $\widehat{i}$ of the atmospheric mesh.

Over the floodplains, the water in the flooded area is able to evaporate at its potential rate. The potential rate of evaporation is defined from the characteristics of the land surface variables of the grid cell to which the HTU belongs. In ORCHIDEE, the transpiration and the interception loss, which are equal to the $\beta$ fraction of the potential evaporation ($E_{pot,bulk}$) with $\beta$ the moisture availability function of the element considered meanwhile the potential evaporation ($E_{pot}$) is used for bare soil and open water evaporation (Barella-Ortiz et al., 2013). Over the floodplains, floodplains evaporation includes the fact that transpiration and interception loss of the vegetation are already calculated by removing their corresponding $\beta$ moisture availability (cf. equation 2).

$$E_{f,i} = f_i(1 - \beta_{vegetation} - \beta_{interception})E_{pot,\widehat{i}} \tag{2}$$

With:

- $E_{pot,\widehat{i}}$: potential evaporation rate over the grid cell $\widehat{i}$

- $\beta_{vegetation}$: $\beta$ coefficient of vegetation

- $\beta_{interception}$: $\beta$ coefficient of interception

The water in the floodplains reservoir is able to infiltrate. It is a one-way flux from the floodplains to the soil moisture of the grid cell. The infiltration term is calculated based on the averaged conductivity for saturated infiltration in the litter layer ($k_{litt}$ in $kg/m^2/s$). This $k_{litt}$ parameter has been established for the soil infiltration processes but not specifically for floodplains. Therefore, the infiltration can be larger than what occurs over the floodplains. This is why a reduction factor ($C$) has been introduced to reduce the floodplains infiltration if necessary. This parameter may depend on the local properties of the region considered such as the type of vegetation or the soil and the sediments which cannot be represented explicitly.

$$I_{f,i} = S_{f,i}.k_{litt}.C \tag{3}$$

## 2.2 Representing the water flow on a graph

Each HTU contains four water reservoirs used by the river routing scheme to represent processes with different time constants: the stream reservoir for the river flow processes, the fast reservoir receiving the surface runoff, the slow reservoir which receives the deep drainage and the floodplain reservoir. The local properties of the HTUs are defined by the elevation change and river length, i.e. the tortuosity of the river segment, aggregated within the HTU. These properties and the characteristics of the water

 

reservoirs govern the residence times of the water in the HTU and thus governs the residence time of the water within the vertex.

For instance, the discharge from the reservoir $j$ of the HTU $i$ ($Q_{j,i}$) is expressed in equation 4 depends on the time constant of the reservoir ($\tau_j$ in s/km) and the topographic index ($topoindex$). The latter is a geographic parameter depending on the slope and the length of the river to define the speed of the water flow and is defined for each reservoir of each HTU ($\alpha_{i,j}$ in km). There are two different $topoindex$: (1) one based on the properties of the pixels composing the HTU which is used for the slow and fast reservoirs and (2) another one based on the properties of the main river of the HTU which is used for the stream and floodplains reservoirs. The time constant of the floodplains ($\tau_f$) is slower than the stream reservoir time constant ($\tau_{stream}$) and faster than the fast reservoir time constant.

$$Q_{x,i} = \frac{V_{X,i}}{\tau_X * \alpha_{i,X}} \qquad \text{with } X \in \{\text{stream}, \text{fast}, \text{slow}, \text{floodplains}\} \tag{4}$$

## 2.3 Water Continuity Equation

### 2.3.1 Stream reservoir

The slow, fast and stream reservoirs are active in all HTUs of the ORCHIDEE routing whether the floodplains are activated or not. However, the floodplains scheme will only impact the functioning of the stream reservoir where a non zero floodplain fraction exists. For this reason, the slow and fast reservoirs will not be mentioned in this section and the common topoindex for the stream and floodplains reservoirs of an HTU i will further be directly noted $\alpha_i$ ($=\alpha_i, stream = \alpha_i, floodplains$).

The water continuity equation provides the basis for the time evolution of the water volumes in the floodplain reservoir. In Figure 1, the different components of the water continuity equation in the case of an HTU with floodplains (Figure 1.a) and without floodplains (Figure 1.b) are displayed.

The volume of water in the stream reservoir of an HTU $i$ ($V_{stream,i}$) follows the water continuity equations in equations 5 differentiating whether it is a HTU with or without floodplains.

$$\frac{\partial V_{stream,i}}{\partial t} = \begin{cases} \sum_{j \in \{i-1/2\}}(F_{out,j}) - F_{out,i+1/2} & \text{if } S_{fmax,i} = 0 \\ Q_{f,i} - F_{out,i+1/2} & \text{if } S_{fmax,i} > 0 \end{cases} \tag{5}$$

With:

- $F_{out,j}$ with $j \in \{i-1/2\}$: Water flowing from the stream reservoir of the upstream HTUs to the HTU $i$.

- $F_{out,i+1/2}$: Outflow from the stream reservoir of HTU $i$ into the stream reservoir of the downstream HTU $i+1$.

- $Q_{f,i}$: Water flowing from the floodplains reservoir of the HTU $i$ to the stream reservoir of the HTU $i$. This variable will be explained in the following subsection.



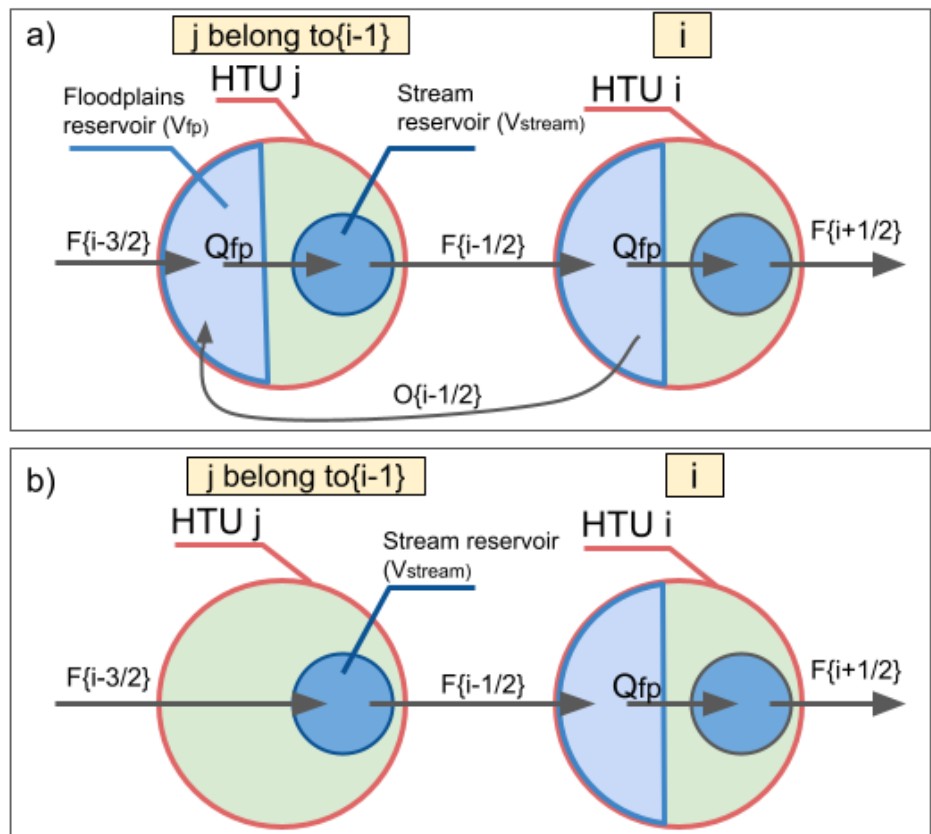

**Figure 1.** Scheme resuming the movement between the different reservoirs for a HTU which has floodplains and its upstream HTUs if (a) the upstream HTU has floodplains or if (b) the upstream HTU doesn't have floodplains.

The outflow from the stream reservoir $F_{out,i+1/2}$ is also affected by the presence of floodplains though a reduction factor based on the fraction of the HTU. The more the HTU is flooded the more the flow out of the stream reservoir is reduced.

This factor aims to represent the impact of the floodplains on the reduction of the river discharge. The floodplains reservoir has its own time constant, therefore, this factor is exclusively used for the stream reservoir. Due to the HTUs structure, some small HTUs over the main river can have a flooded fraction close to 1 that impedes the river from flowing and a parameter $R_{limit}$, equal for all HTUs, has been implemented to limit this flow reduction. This parameter is the same for all the HTUs. The reduction factor can be deactivated with a value of $R_{limit} = 0$. Therefore, the formulation of the outflow from the stream

reservoir of an HTU $i$ which has floodplains ($F_{out,i+1/2}$) differs from the equation 4 and is represented in equation 6.

$$F_{out,i+1/2} = \frac{V_{f,i}}{\tau_{stream} * \alpha_i} * (1 - max(f_i, R_{limit})) \qquad (6)$$





The flooded fraction $f_i$ used in equation 6 is calculated from the area of the HTU which is flooded ($S_{f,i}$). This value is diagnosed using the equation 7.

$$S_{f,i} = min(\Gamma(V_{f,i}), S_{fmax,i}) \tag{7}$$

The appropriate function $\Gamma$ will be discussed further in Subsection 2.4.

### 2.3.2   Floodplains reservoir

This subsection focuses on the definition of the different water fluxes related to the floodplains reservoir. The water continuity equation governing the temporal changes of the volume of water in the floodplains reservoir ($V_{f,i}$) is presented in equation 8. The different components of this equation will be described further in this subsection.

$$\frac{\partial V_{f,i}}{\partial t} = (P_{f,i} - E_{f,i} - I_{f,i}) - Q_f + \sum_{j \in \{i-1/2\}} (F_{out,j} - O_j) \tag{8}$$

With:

- $F_{out,j}, j \in \{i-1/2\}$: Water inflow into the floodplains from the upstream HTUs,

- $O_j, j \in \{i-1/2\}$: Overflow of HTU $i$ into the floodplains reservoir of the upstream HTUs.

- $P_{f,i}$: Rainfall onto the floodplain,

- $E_{f,i}$: Evaporation from the flooded surface,

- $I_{f,i}$: Infiltration from the floodplain into the soil moisture reservoir,

Within a HTU $i$ with floodplains, the flow of water from the floodplains reservoir to the stream reservoir ($Q_{f,i}$) has the same type of formulation as equation 4. This formulation is presented in equation 9.

$$Q_{f,i} = \frac{V_{f,i}}{\tau_f * \alpha_i} \tag{9}$$

The floodplains scheme allows a specific HTU to "overflow" the content of its floodplains reservoir into connected upstream HTUs with floodplains. This process is driven by the difference in height between the elevation of the water and that of the neighbouring HTUs:

$$\Delta h_{i,j \in \{i-1/2\}} = max((z_i + h_i) - (z_j + h_j), 0) \tag{10}$$

with:





– $z$: the elevation at the outflow of the HTU

           – $h$: the water level in the floodplains

As long as $\Delta h_{i,j \in \{i-1/2\}} = 0$, there is no overflow ($O_{j \in \{i-1/2\}} = 0$). When the water rises over the elevations of the upstream HTU, the overflow is enabled. The flux is proposed to be:

$$O_{j \in \{i-1/2\}} = \Delta h_{i,j} \frac{S_{f,i} S_{f,j}}{S_{f,i} + S_{f,j}} \frac{1}{OF} \tag{11}$$

with $OF$ the time constant of the overflow (in days).

At the edge of the atmospheric grid cell, some small HTUs are created due to the overlap between the catchments and the grid cells. These HTUs may generate numerical issues such as unrealistically high $\Delta h$ values due to their small area. The volume of water which overflow from an HTU to its upstream HTU is calculated from the excessive floodplains height $\Delta h$ and a surface. In order to solve the undesirable numerical effects, both the surface of the HTU which overflows and of its upstream

HTU are considered using the following surface: the term $\frac{S_{f,i} S_{f,j}}{S_{f,i} + S_{f,j}}$.

Excessively low values of the $OF$ time constant are another source of numerical instabilities (the lower $OF$, the more important the overflow). For example, if the HTU $i$ overflows in various upstream HTUs, an excessive transfer of water at once will leave a negative volume in the floodplains reservoir which generates an oscillation between HTU outflow and downstream overflow. It is a time step issue which depends on the choice of OF relative to the time step of the scheme. It is possible

to increase the overflow without generating instabilities by using a time-splitting scheme to solve this, i.e. by repeating the overflow operation several times during the same time step using a slower time constant $OF$. The number of repetitions of the overflow water transfer within a single time step is defined by the parameter $OF_{repeat}$.

## 2.4  Floodplains geometry

Another crucial aspect of the floodplains scheme is the relationship between the volume of water in the floodplain reservoir
($V_{f,i}$), the surface of open water and the height of water. In order to establish a simple but meaningful relationship, some assumptions about the geometry of the floodplains are necessary.

As the HTUs are constructed from higher resolution hydrological data, it is possible to derive a direct relationship using the topography data from the hydrological pixels (Dadson et al., 2010; Zhou et al., 2021; Fleischmann et al., 2021; Chaney et al., 2020). But this method would bring two different issues: (1) the uncertainty of the topography over lowlands such

as floodplains and (2) the high computational memory cost. The memory cost involved may not necessarily be worth the improvement it would bring to the simulation.

For this reason, the definition of the floodplains shape has been simplified by using two variables controlling the shape of the floodplains such as proposed by D'Orgeval (2006) and shown in Supplementary Figure B2.a. These variables are the following:

           – $h_{0,i}$: the height at which the floodplains of the HTU reach their full extension, i.e. $S_{f,i} = S_{fmax}$.





– $\beta_i$: the shape of the floodplain which will control how quickly it fills. $\beta_i > 1$ corresponds to a floodplain with a concave cross section (as Supplementary Figure B2.b) whereas $\beta_i < 1$ corresponds to a floodplain with a convex cross section. $\beta_i = 1$ represents a triangular cross section.

In D'Orgeval (2006), both variables have been set to constant values: $\beta = 2$ and $h_0 = 2m$. With the high resolution flood-plains scheme, it is possible to define $\beta$ and $h_0$ with more precisely using the characteristics of the HDEM pixels combined

within an HTU. This is described in 2.4.3.

The spatial representation of the floodplains in an HTU $i$ is defined by the relationship between the volume of water in the floodplains $V_{f,i}$, the surface of the floodplains $S_{f,i}$ and the height of the floodplains $h_i$. It is considered that at a certain height $h_0$, the whole floodplain is flooded, i.e. $S_f = S_{fmax}$ and that, even if the floodplains height is higher than $h_0$, the flooded area cannot exceeds this limits (cf. equation 12). The shape of the floodplains will have an influence only for $h_i < h_0$ because above

$h_0$, the height is considered to increase linearly with the volume.

$$f_i = \frac{\min(S_{f,i}, S_{fmax,i})}{S_i} \tag{12}$$

### 2.4.1    Case $S_{f,i} < S_{fmax,i}$

If we consider a HTU $i$ which has a potential flooded area of 100%, i.e. with $f_{max,i} = 1$ or $S_{fmax,i} = S_i$, the relationship between the flooded area $S_{f,i}$ and the height of the floodplains $h_i$ for $h_i < h_0$ is represented in equation 13.

$$S_{f,i} = S_{B,i} \left( \frac{h_i}{h_{0,i}} \right)^{\beta_i} \tag{13}$$

This assumes that the transect of the floodplain has an exponential shape and with the choice of $\beta$ it can be decided how quickly it fills. The relation between the floodplains height and the volume in the floodplains reservoir is obtained by integrating

this function between 0 and $h_i$ yielding:

$$V_{f,i} = \frac{S_{B,i}}{\beta_i + 1} \frac{h_i^{\beta_i + 1}}{h_{0,i}^{\beta_i}} \tag{14}$$

This provides the $\Gamma$ function introduced above to calculate the surface from the volume. The above equations are only valid for $h \leq h_{0,i}$.





### 2.4.2 Case $S_{f,i} = S_{fmax,i}$

If $h_i > h_{0,i}$, we have $S_{f,i} = S_{fmax,i}$. The equation 15 shows the relationship between the flooded surface and the volume in the floodplains reservoir by combining the equations 13 and 14.

$$S_{f,i} = \Gamma(V_{f,i}) = \max(\frac{S_{B,i}}{h_{0,i}^{\beta_i}} \left[ \frac{(\beta_i + 1)h_{0,i}^{\beta_i} V_{f,i}}{S_{B,i}} \right]^{\frac{\beta_i}{\beta_i + 1}}, S_{fmax,i}) \tag{15}$$

In order to generalize, the floodplains height above $h_0$ increases linearly with the volume. Considering $V_{fmax,i}$ the volume at which $\Gamma(V_{fmax,i}) = S_{fmax,i}$. For $V_{f,i} > V_{fmax,i}$, the flooded surface and the floodplains height in the HTU $i$ follows respectively the equation 16 and 17.

$$S_{f,i} = S_{fmax,i} \tag{16}$$

$$h_i = h_0 + \frac{(V_{f,i} - V_{fmax,i})}{S_{fmax,i}} \tag{17}$$

### 2.4.3 Orography and shape of the floodplains

The elevation is a variable available for each pixel in the HDEM. Considering a HTU $i$, the reference elevation is defined by the elevation of the outflow pixel ($z_i$) meanwhile $h_{0,i}$ is the lowest difference of elevation between $z_i$ and its upstream HTUs reference elevation (cf. equation 18).

$$h_{0,i} = \min_{j \in \{i-1\}}(z_j - z_i) \tag{18}$$

The $\beta$ variable has been estimated using the standard deviation of the distribution of the elevation including the values for all the HDEM pixels within the HTU. The different values of standard deviation are bounded by $\mathrm{lowlim\_std} = 0.05m$ and $\mathrm{uplim\_std} = 20m$ and are then converted to obtain the $\beta$ variable which ranges between values of $\mathrm{lowlim\_beta} = 0.5$ and $\mathrm{uplim\_beta} = 2$.

$$\mathrm{std\_orog\_bounded}(i) = \begin{cases} \mathrm{lowlim\_std} & \text{if } \mathrm{std\_orog}(i) < \mathrm{lowlim\_std} \\ \mathrm{uplim\_std} & \text{if } \mathrm{std\_orog}(i) > \mathrm{uplim\_std} \\ \mathrm{std\_orog}(i) & \text{elsewhere} \end{cases} \tag{19}$$




$$\beta_i = \frac{\text{std\_orog\_bounded(i)}}{\text{uplim\_std} - \text{lowlim\_std}}(\text{uplim\_beta} - \text{lowlim\_beta}) \tag{20}$$

With the hypothesis that $h_{0,i}$ is the height at which the floodplain of the HTU $i$ is totally flooded, this height is assumed to be the minimum of the difference between elevation of the HTU $i$ and the ensemble of its inflows that have floodplains ($\{i-1\}$). When $h_i$ is larger than this difference, it means that the floodplain of $i$ will be able to overflow to the upstream vertices.

The conversion of water volume in the floodplains reservoir into an open water area has been assessed by testing different values of the default parameters defining the floodplains shape ($\beta$ and $h_0$). It results that, although these parameters can lead to important changes over a single HTU, they have a limited influence on the total flooded area over a larger region (not shown).

## 2.5   Ancillary data

### ORCHIDEE's high resolution routing

The routing in ORCHIDEE has been constructed by the routing preprocessor (RoutingPP) presented in (Polcher et al., 2022). It allows to combine different high resolution hydrological information to construct the HTUs and calculate their characteristics. In this case, the routing graph have been constructed using the MERIT-Hydro dataset.

### Spatial description of the floodplains

The Global Lake and Wetland Database (GLWD -  WWF, 2004) available at a 1 km resolution has been interpolated to the HDEM used in order to define a mask of potentially flooded areas based on the following categories: (1) Freshwater Marsh, Floodplains; (2) Reservoir; (3) Pan, Brackish Saline Wetland. Therefore, the floodplains mask is available for each pixel of the hydrological data. This data allows to calculate a potentially flooded area for each HTU during the routing construction.

The floodplains map in GLWD covers all the Pantanal region. Before using the floodplains scheme over other large flood-
plains, it is necessary to assess the relevance of the spatial extent of the categories considered as floodplains in GLWD. In case of inadequacy of the GLWD representation over the region, other datasets can be used to define the potentially flooded area.

### Calibration of the parameters

The different parameters of the floodplains scheme have been calibrated based on the simulated discharge at the *Porto*
*Murtinho* station, which is the reference station at the outflow of the Pantanal (Brazil, lat: 21.7°S, lon: 57.9°W) between 1991 and 1996 in comparison to the observations considering: (1) the variation of the discharge through its correlation with the observations and (2) the mean value and variability of the discharge. The parameters have been adjusted in the following order :

    1. $\tau_f$ and $R_{limit}$: the floodplains reservoir time constant and the discharge limiter which have the greater influence on
discharge (mean value and variability). The interval considered for $\alpha_i$ is $[\alpha_{stream}, \alpha_{fast}]$ because the flow from the floodplains is considered to be slower than the stream reservoir but faster than the fast reservoir. The interval $[0,1]$



was searched for an optimal value of $R_{limit}$. The best combination of parameters has been established through a grid-search method which consists in evaluating the different combinations of parameters within their respective interval of definition.

2. $OF$ and $OF_{repeat}$: these parameters slightly influence the temporality of the discharge. The interval considered for $OF$ was [0.5 day, 2 days] only and for $OF_{repeat}$ the interval [1,5 repetitions] was searched.

    3. $C$: the infiltration constant which determines the loss to soil moisture and, thus, potentially to evaporation. It is therefore able to reduce / increase the level of the discharge at the outflow of the region. The interval considered for $C$ is [0,1].

The values of the parameters found depended on the resolution of the atmospheric forcing and are shown in Table 1. The
parameterization for the 0.5° resolution has been established with WFDEI_GPCC atmospheric data forcing and the 20 km resolution with AmSud_GPCC atmospheric data forcing which are both described in more details in the following section. It is recommended to make a sensitivity test before using the scheme over another region to evaluate if this parameterization is the more appropriate.

| Resolution | 0.5° | 20 km |
|---|---|---|
| $\tau_f$ [s/km] | 15 | 20 |
| $R_{limit}$ [-] | 0.4 | 0.4 |
| $OF$ [day] | 1 | 1 |
| $OF_{repeat}$ [-] | 3 | 3 |
| $C$ [-] | 0.7 | 1 |

**Table 1.** Parameterization of the floodplains scheme depending on the resolution of the atmospheric grid.

## 3 Methodology and Dataset

### 3.1 Methodology of Validation and Analysis

Two pairs of ORCHIDEE simulation using the high resolution routing (HR) are used to perform the validation of the floodplains scheme and the analysis of the impact of the floodplains on ORCHIDEE. Each pair is forced by a different atmospheric forcing: WFDEI_GPCC at 0.5° resolution and AmSud_GPCC at 20 km resolution and is composed by a simulation with the floodplains scheme activated (FP) and another one without the floodplains scheme (NOFP). The use of two forcings with
different resolutions allows us to assess the influence of the resolution and the forcing uncertainty on the floodplains scheme. The forcings are further described in Subsection 3.3.

The analysis is performed between 1990 and 2013 which is the period over which both forcing data sets are available. The following validation and analysis of the simulation will focus on the mean values and annual cycle of the variables between





1990 and 2013 but will also consider their mean values over different seasons during this period: the Flood Season from March
to May (MAM) and the Dry Season from September to November (SON).

## 3.2    Model Description: ORCHIDEE

The simulations presented in this publication are the output of off-line ORCHIDEE simulations, i.e. simulations of the OR-
CHIDEE LSM forced by an external atmospheric dataset containing the atmospheric data required to run the model (downward
long and shortwave radiations, precipitation, 2-m air temperature, wind speed, 2-m specific humidity, snowfall and rainfall).
Ancillary data-sets provide the information about vegetation cover and the soil composition.

The soil properties is described by the combination of the three main soil textures: coarse, medium and fine from the USDA
soil description (Reynolds et al., 2000).

The vegetation in ORCHIDEE is described in the model's input by the potential vegetation cover (maxvegetfrac) for 12
different Plant Function Types (PFTs) and the fraction of bare soil cover. These bare soil can be covered by different types of
non-vegetated land surfaces such as glaciers, cities lakes or flooded areas. For each grid cell, the sum of the maxvegetfrac of
the different PFTs and of the bare soil surfaces is equal to 1. In these simulations, the PFTs are constructed from the ESA-CCI
data-base, (European Space Agency-Climate Change Initiative; ESA, 2017). Reader should be aware that original ESA-CCI
remote PFTs classification has been post-processed to the 13 ones used in ORCHIDEE.

The vegetation cover is defined by the fraction of the grid cell occupied by each PFT (vegetfrac) whose upper limit is
maxvegetfrac. It is driven by the Leaf Area Index (LAI, in $m^2/m^2$) of the PFT, if LAI $\geq$ 1 then vegetfrac = maxvegetfrac
elsewhere the fraction not covered by this vegetation type is considered by the model as bare soil.

The potential vegetation cover used in these simulations is shown in Supplementary Figure B1. It shows maxvegetfrac
over the region for the different PFTs categories existing over the Pantanal. There is a high presence of Tropical Broadleaf
Evergreen on the Western and Northeastern part of the Pantanal covering more than 50% of the grid cells in this region. The
Tropical Broadleaf Raingreen is present over all the Pantanal with a cover of around 20% of each grid cells. The rest of the
Pantanal is mainly covered by Natural Grassland of C3 (in the Northwest, the South and the East) and C4 type (in the North
and the South/Southeast).

The hydrology in ORCHIDEE is represented through a 11-layer soil scheme (De Rosnay et al., 2000; de Rosnay et al., 2002;
Campoy et al., 2013) representing the vertical movement of the water in the soil and the transfer of heat.

The surface energy budget is the partitioning of the total net radiation composed by the net longwave and shortwave radi-
ations ($Rn = LW_n + SW_n$) into latent heat fluxes (LE), sensible heat fluxes (H) and ground heat fluxes (G), cf. eq. 21. The
net shortwave radiation is determined by the albedo ($\alpha$) at the surface because $SW_n = (1 - \alpha)SW_{in}$ with $SW_{in}$ the incom-
ing shortwave radiation. In the model, the impact of the flooded area on the albedo is not considered, but the changes in soil
moisture and vegetation directly affect this parameter and, therefore, impact the net radiation.

$$R_n = LE + H + G \tag{21}$$



The latent heat flux is represented by the latent heat of vaporization ($L$) and the evapotranspiration ($E$). The sensible heat fluxes ($H$) in the ORCHIDEE LSM are driven by the difference between the surface temperature ($T_s$) and the temperature of the air at the surface ($T_a$). It is calculated from the equation 22 with $c_p$ the specific heat.

$$H = \frac{\rho c_p}{r_a} [T_s - T_a] \tag{22}$$

Over a large period of time, the ground heat fluxes can be neglected and, then, $R_n$ is only partitioned into LE and H (G=0). Thus, the relative distribution of LE and H is important to quantify the changes in the surface energy budget and the changes in temperature. This can be expressed with the Evaporative Fraction (EF) which is the ratio of latent heat (LE) over the total land atmosphere fluxes, i.e. the sum of the latent heat and of the sensible heat (LE+H):

$$EF = \frac{LE}{LE + H} \tag{23}$$

The EF index gives an indication of the distribution of the heat fluxes over land. The value of this index tends to 0 when there are no latent heat fluxes such as in arid areas. It can take the value of 1 if there are only latent fluxes and take values over 1 when the land surface is cooled because in this case $H < 0$.

### 3.3 Forcings

WFDEI_GPCC is a 0.5° resolution atmsopheric forcing data set for land surface models (Weedon et al., 2014). It is derived
from the ERA-Interim reanalysis processed by the WATCH Forcing Data methodology (Dee et al., 2011) and has a temporal resolution of 3 hours and a spatial discretization of 0.5°. WFDEI_GPCC corresponds to the version of WFDEI whose precipitation has been bias-corrected by the GPCC dataset (Schneider et al., 2017).

AmSud_GPCC is a 20km resolution forcing based on the bias-corrected AmSud simulation, a 30 years simulation performed with the RegIPSL regional model (Guion et al., 2022) from 1990 to 2019 and forced with ERA5 re-analysis data. The precipi-
tation of the AmSud simulation has been bias-corrected by the GPCC monthly precipitation adjusting the monthly precipitation total by a multiplicative factor for each grid cell to obtain the AmSud_GPCC forcing. This has been done to correct the negative biases of precipitation over the Southern Amazon and Northern La Plata Basin (i.e. the Upper Paraguay River Basin).

It must be emphasized that none of these two forcings includes the impact of the floodplains and, thus, includes large biases in lower atmospheric temperature and humidity over this region.
As ORCHIDEE is used in off-line mode, the atmospheric conditions are fixed and the floodplain parameterization does not interact with the atmosphere and doesn't affect the atmospheric conditions. Therefore, these forcings will be a source of errors for the near surface temperature and humidity because they will not respond to the changes related to the presence of flooded areas in the model.

It should also be noticed that, compared to WFDEI_GPCC, AmSud_GPCC has a higher evaporative demand due to overes-
timated near-surface temperature, incoming shortwave radiation and an underestimation of near surface humidity (cf. Supplementary Figure B3). This is partly related to the fact that the AmSud_GPCC forcing is only bias corrected as to the precipitation,



the other variables remain unchanged. Therefore, as the precipitation is underestimated over the region in AmSud, the other variables represent a drier atmosphere over the Pantanal.

### 3.4 Discharge

The National Hydro-meteorological Network managed by the Brazilian National Water Agency (Agência Nacional de Águas - ANA) has provided the monthly river discharge observations for the Porto Murtinho station.

This station is considered as the reference outflow for the Pantanal (Schrapffer et al., 2020; Penatti et al., 2015). Moreover, its large continuous data record allows for choosing freely the period of simulation to evaluate the floodplains scheme.

### 3.5 Flooded area

Depending on the period simulated, the flooded area simulated was assessed by different estimates of the flooded area over the Pantanal.

The evolution of the flooded area over the 20$^{th}$ century has been estimated by Hamilton et al. (1996) and Hamilton (2002) extrapolating the correlation between river height and the flooded area established over the period 1979 and 1987.

Padovani (2010) performed a satellite estimate of the flooded area by applying a Linear Spectral Mixture Model to MODIS
data between 2002 and 2009.

Apart from Hamilton (2002) and Padovani (2010), the satellite estimate of the flooded area based on the modified Normalized Difference Water Index (mNDWI) index using the normalized difference between green and Short-Wave Infrared bands presented in Schrapffer et al. (2022, in press) is also used to assess the flooded area in the FP simulations.

The Global Inundation Extent from Multi-Satellites database version 2 (GIEMS-2; Prigent et al., 2020) is a satellite estimate
of flooded areas (agricultural irrigation and wetlands) which is principally constructed using passive microwaves observations but also using visible and near-infrared reflectance data from optical satellites. GIEMS-2 is a global monthly dataset available at a 0.25° resolution between 1992 and 2015. As for the GIEMS version 1, the GIEMS-2 is largely used to validate the floodplains representation in different models (Zhou et al., 2021; Marthews et al., 2021).

### 3.6 Water Mass

In ORCHIDEE Total Water Storage (TWS) is defined by summing the different reservoirs of the routing scheme (slow, fast, stream and floodplains) and the soil moisture, in order to obtain an estimate comparable to the water storage from the GRACE satellite (Ngo-Duc et al., 2007).

The Gravity Recovery And Climate Experiment (GRACE; Schmidt et al., 2008)) satellite mission is an US-German collaboration launched in March 2002. The GRACE twin satellite aims to estimate the changes of the mass redistribution near the
surface which are related to different processes by evaluating the changes of the gravity fields. GRACE data represents the anomaly of water mass normalized by the values obtained during the 2004-2010 period. The data from GRACE is available since 2002.





In order to reduce the signal to noise ratio, it is recommended to use the GRACE data at spatial scale of 90000 km$^2$
(Vishwakarma et al., 2021). The extension of the Pantanal of approximately 150 000 km$^2$ (Barbosa da Silva et al., 2020) is

large enough to be able to use GRACE.

## 4   Validation of the simulated floodplains

### 4.1   Discharge

The annual cycle of the discharge between 1990 and 2013 is shown in Figure 2. The activation of the floodplains scheme
improves the seasonality of the annual cycle with a peak in July in the FP simulations as in the observations instead of February

in the NOFP simulations. The mean annual discharge and the amplitude of the discharge are also reduced in the FP simulations
compared to NOFP and are therefore in better agreement with observations. This can directly be explained by the loss of water
from the river system to the soil moisture (floodplains infiltration) and to the direct evaporation from the floodplain. However,
the amplitude of discharge simulated in FP is still overestimated with higher discharge during the peak and lower discharge
between November and February. This discharge is more important in WFDEI_GPCC_FP than AmSud_GPCC_FP.

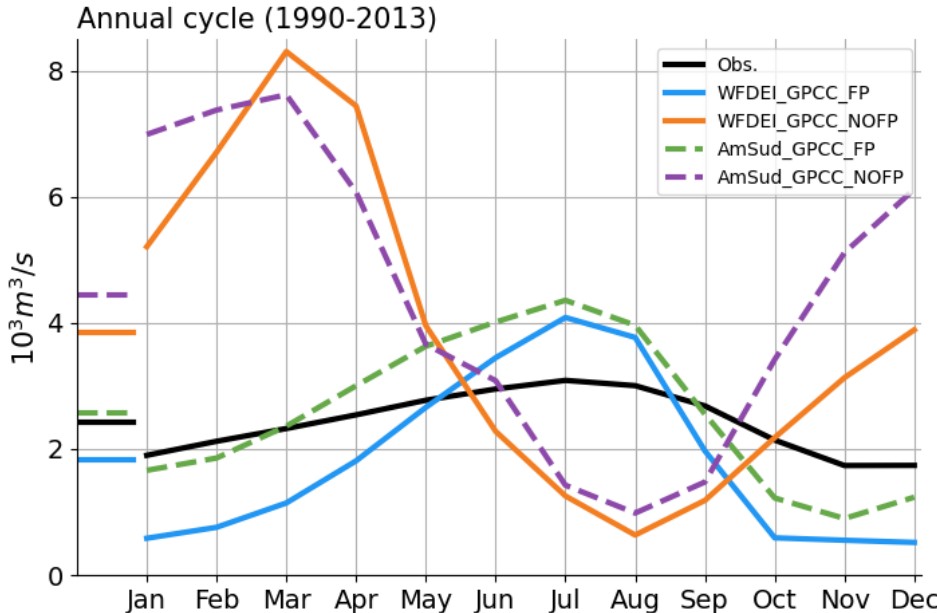

**Figure 2.** Annual cycle of the discharge at the Porto Murtinho station between 1990 and 2013 for the simulations with (blue) and without
(red) the floodplains scheme activated for the forcing WFDEI_GPCC (solid line) AmSud_GPCC (dashed line) compared to the observations
(black line). The mean annual discharge is represented by a horizontal line on the left.





The statistical indexes calculated to summarize the analysis are presented in Table 2 and are described in the Supplementary material A. The activation of the floodplains scheme leads to a substantial improvement of the simulations with higher values of the correlation and of the Nash–Sutcliffe model efficiency coefficient (NSE) while the Root Mean Square Error (RMSE) is closer to 0 and the Percent Bias (PBIAS) is lower. The correlation with observations of the simulated discharge in the NOFP simulations are not significant. WFDEI_GPCC_FP has a better correlation and NSE than AmSud_GPCC_FP while the

opposite is true for the PBIAS and the RMSE. Based on the previous analysis, WFDEI_GPCC_FP seems to better represent the annual cycle of discharge compared to AmSud_GPCC_FP which results in higher correlation and NSE but, as the amplitude of the annual cycle is higher than in the observations, its RMSE and a PBIAS are worse than the values for AmSud_GPCC_FP.

    Considering the dry atmospheric bias and thus higher potential evaporation in AmSud_GPCC compared to WFDEI_GPCC and the fact that both forcings have a similar precipitation, we may expect the discharge in AmSud_GPCC_FP to be lower than

the discharge in WFDEI_GPCC_FP, however, the opposite is true. This suggests that the resolution of the interactions with the atmosphere may be playing a role in the representation of the floodplain processes and, therefore, in the water cycle of the basin. For the same floodplains scheme, the coarser resolution has difficulties representing the low flows and it results in a strong overestimation of the difference between the high and low value of discharge compared to the observations. The simulations without floodplains (WFDEI_GPCC_NOFP and AmSud_GPCC_NOFP) have similar low flows and variability. From Polcher

et al. (2022), we know that an increased number of HTUs does not change the simulation of the discharge. Therefore, we can conclude that the effect of the floodplains on the hydrology seems to be better captured by the high resolution simulation.

| Forcing | NSE | PBIAS (%) | RMSE ($m^3/s$) | Corr |
|---|---|---|---|---|
| WFDEI_GPCC_HR_NOFP | -0.10 | 58.46 | 1213.59 | -0.09 |
| WFDEI_GPCC_HR_FP | 0.44 | -24.90 | 448.20 | 0.74* |
| AmSud_GPCC_HR_NOFP | -0.17 | 83.07 | 1321.88 | -0.31 |
| AmSud_GPCC_HR_FP | 0.36 | 5.53 | 383.30 | 0.60* |

**Table 2.** Evaluation of the discharge at the outflow of the Pantanal for the simulations with the high resolution routing scheme with and without the floodplains scheme activated forced by two atmospheric forcings with a different resolution (WFDEI_GPCC and AmSud_GPCC) using statistical index (NSE, PBIAS, RMSE, Corr).

## 4.2   Water mass

The water mass in the WFDEI_GPCC and AmSud_GPCC pairs of simulations is analyzed in this subsection to help understand the dynamics of the model in its representation of the water cycle at different resolutions.

The evolution of the monthly total water mass anomaly in the simulations normalized by the 2004-2010 mean values can be compared to GRACE over the Pantanal region. Due to its resolution, GRACE is a coarse estimate but it can provide a general overview and qualitative evaluation of the representation of the water cycle in the model. Therefore, the area considered to



calculate the anomaly of the normalized Total Water Storage for GRACE and the simulation is a rectangle which goes from 61 to 53°W and from 15 to 21°S. It includes the Pantanal which represents a third of the total area over this rectangle.

Table 3 shows the correlation between the total water mass anomaly from GRACE and from the simulations. The high level of correlation shows that all the simulations show an annual evolution similar to that observed by GRACE. However, the small differences between the FP and the NOFP simulations for both forcings has to be noted. This means that the model is properly representing the evolution of the water volume in the reservoirs over the Pantanal but that the floodplains reservoirs have little impact on the Total Water Storage.

| Forcing | Correlation |
| --- | --- |
| WFDEI_GPCC_NOFP | 0.951* |
| WFDEI_GPCC_FP | 0.958* |
| AmSud_GPCC_NOFP | 0.711* |
| AmSud_GPCC_FP | 0.701* |

**Table 3.** Correlation between the anomaly of total water storage from GRACE and the volume of water in the different reservoir over the Pantanal region between 2003 and 2013 normalized by the mean and standard value between 2004 and 2010. The asterisk signals that the correlation has a level of significance higher than 99%.

Figure 3.a compares the contribution of different reservoirs in the model over the Pantanal through the annual mean of the water volume within each one of them. The floodplains scheme has a similar impact on the reservoirs at both resolutions. The volume of water in the soil moisture and in the stream reservoir increases slightly when activating the floodplains scheme. This increase is even more important in the fast and slow reservoirs. The activation of the floodplains allows to store more water in the river network when the floodplains scheme is activated.

The annual cycle of the water volume within the different reservoirs of the model over the Pantanal is shown in Figures 3.b-f. The soil moisture over the Pantanal is the largest contribution to the Total Water Storage followed by the stream reservoir and the floodplains reservoir in the FP simulations. The activation of the floodplains increases soil moisture due to the infiltration of the water from the floodplains reservoir but is not significantly impacting its temporal evolution (cf. Figure 3.b). This explains why the floodplains scheme doesn't have an impact on the correlation of the total water storage over the Pantanal. Concerning

the stream reservoir (cf. Figure 3.c), we observe a change that is similar to the change of the discharge at the Porto Murtinho station because this is the reservoir that drives the discharge. The floodplains reservoir (cf. Figure 3.d) has logically a value of 0 in the NOFP simulation and follows the evolution of the stream reservoir in the FP simulations because these reservoirs are connected. The content of water in the slow reservoir (cf. Figure 3.e) plays, in the model, the role of the aquifers. Its volume strongly increases due to the floodplains and this is related to increased deep drainage induced by the higher soil moisture

infiltration when the floodplains are activated (cf. Figure 3.b). The water content in the fast reservoir (cf. Figure 3.f) displays much lower volumes compared to the other reservoirs. However, it is higher in the FP simulation compared to the NOFP simulation. Despite the fact that soil moisture is higher in WFDEI_FP, AmSud_FP seems to have more runoff. This can be related to the fact that the area of saturated soil moisture over the floodplains is larger in WFDEI_FP.



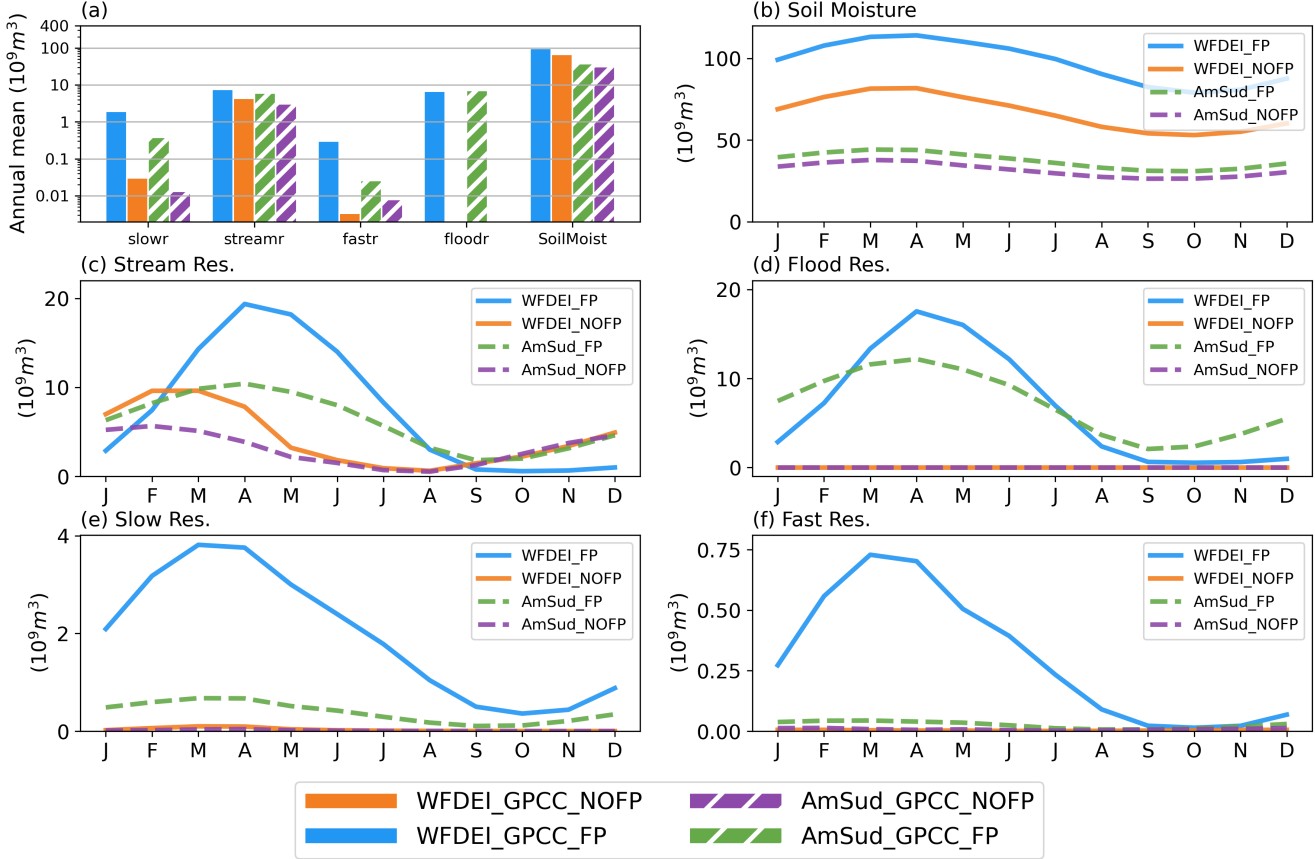

**Figure 3.** Annual cycle of the content of water in the different reservoir: slow (a-b), stream (c-d), fast (e-f), flood (g-h), integrated soil moisture (i,j) for the pair of simulations FP and NOFP forced by WFDEI_GPCC (a,c,d,e,g,i) and AmSud_GPCC (b,d,f,h,j). (k) shows the annual mean value of the water storage in the different reservoirs, a logarithmic scale is used to facilitate the comparison.

The water volumes are higher in WFDEI_GPCC for the fast, slow and stream reservoirs compared to AmSud_GPCC.
This can be related to the higher evapotranspiration in AmSud_GPCC compared to WFDEI_GPCC due to the dry bias in the atmospheric conditions in this forcing. The higher evapotranspiration decreases soil moisture, drainage and runoff. In consequence, the volume in the fast and slow reservoirs also decrease. The stream and flood reservoirs are less affected by the higher evapotranspiration over the Pantanal as their dynamic is more influenced by the water from the upstream areas that flows into the Pantanal. The origin of the overestimated discharge annual variability at Porto Murtinho in WFDEI_GPCC_FP can
attributed to the slow, fast, stream and floodplains reservoirs which have a more pronounced amplitude of their annual cycle than AmSud_GPCC with much higher volume of water in the slow and fast reservoirs.





In conclusion, the floodplains involve relatively small masses of water compared to the total water storage over the region but these volumes are of great importance as they directly affect the river discharge and form open-water surfaces which will impact the land-atmosphere interaction.

## 535  4.3  Flooded Area

The evolution of the simulated flooded area for the FP simulations is presented in Figure 4. In this case, the flooded area is compared to the observationally based estimates over the period 1992-2013 as it allows to compare directly with different satellite derived products: GIEMS-2 (Prigent et al., 2020), Hamilton (2002) and Padovani (2010). The flooded area in WFDEI_GPCC_FP is higher than in AmSud_GPCC in terms of mean value and inter-annual variability. Despite the differ-

ences, the mean value of the flooded area is within a similar range of value in both simulations with the floodplains activated.

There are discrepancies between different satellite estimates considered in this study. Hamilton (2002) tends to estimate higher areas compared to GIEMS-2 while the opposite is true for Padovani and the mNDWI based satellite estimate from Schrapffer et al. (2022, in press). The mean value of the simulated extent seems to be underestimated by the model compared to satellite estimations and correspond to the lowest value of the satellite estimate. The annual variation are correlated but the

variability of the simulated flooded area is strongly underestimated.

GIEMS-2 also allows us to directly compare the observed spatial extent with the simulated area in ORCHIDEE. Figure 5 represents the geographic distribution of the flooded fraction averaged over the 1992-2013 period as well as the temporal correlation and root mean square error between each simulation and GIEMS-2. The structure of the mean flooded fraction in AmSud_GPCC_FP and WFDEI_GPCC_FP is similar to GIEMS-2 (cf. northern region and floods over the main Paraguay

river). The flooded area at the center of the Pantanal is not captured by the model, this is related to the presence of the *Taquari Megafan*, which is an area of divergent flows which very sensitive to the orography and cannot be represented in this model (Louzada et al., 2020; Assine, 2005). Both forcings result in a similar structures of the flooded areas, the simulation using the higher resolution forcing captures the spatial pattern better. The higher resolution in AmSud_GPCC_FP allows to observe the higher concentration of flooded area over the main Paraguay river and the impact of the overflow to the adjacent grid cell. At

higher resolution, the overflow is more important as the HTUs are smaller and, therefore, reaches higher floodplains height over the largest river.

The correlation between time series of flooded areas over the Pantanal in the simulations and GIEMS-2 (Figure 5.c and g) is relatively high in the northern and central regions reaching values higher than 0.6. However, the flooded fraction South of the Pantanal no correlation in the AmSud_GPCC_FP. This may be related to the presence of ponds which are isolated from the

flood pulse of the large rivers flowing through the Pantanal (Nhecolândia ponds, Guerreiro et al., 2019).

The differences between the simulations and GIEMS-2 are also assessed grid point by grid point using the Root Mean Square Error over the region in Figure 5.d and h. Lower values shows a good correspondence of the flooded fraction between the model and GIEMS-2 while the higher values (darker grid points) show larger discrepancies. The major error are related to (1) the Taquari Megafan flooded area not represented in the model, (2) flooded area in the Northeast (*Cuiaba* and *São Lourenço*



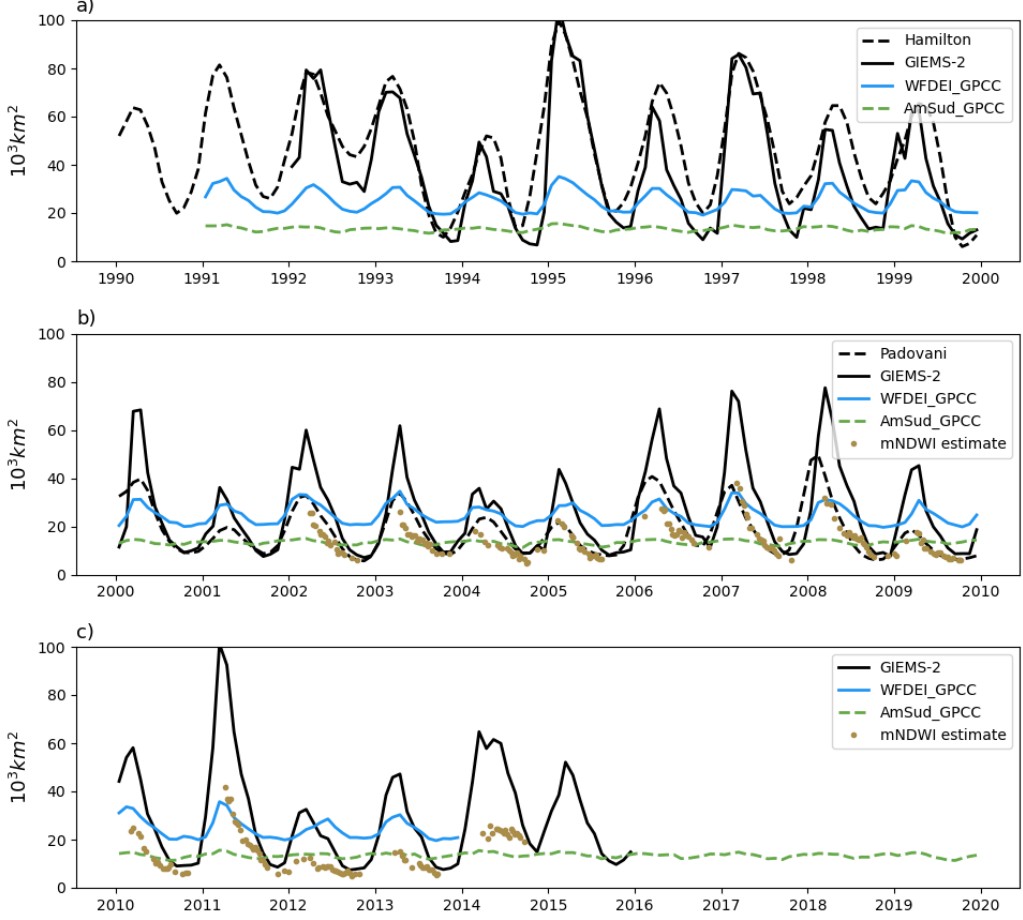

**Figure 4.** Time series of the flooded area in the simulations with the high resolution floodplains scheme (HR) forced by WFDEI_GPCC and AmSud_GPCC for the 1990's (a), 2000's (b) and 2010's (c) in comparison to the different satellite estimate available over the region: Hamilton (2002) until 2000, Padovani (2010) between 2000 and 2010, GIEMS-2 (Prigent et al., 2020) for the period 1992-2015 and the flood estimate based on MODIS MOD09A1 using the mNDWI spectral index (Schrapffer et al., 2022, in press).

river) which are not present in GIEMS-2, (3) the fact that the flooded areas in AmSud_GPCC are concentrated along the main rivers while the flooded areas are more extended in GIEMS-2.

     Although the variability of the floodplains seems to be underestimated in the model, the spatial representation of the flooded area is realistic. The high resolution atmospheric grid allows a more precise description of the flooded area. The underestimation of the variability can be related to: (1) the fact that model handle separately the saturated soils and the flooded area while the

satellites estimates consider them together, (2) the conversion of the volume of water in the floodplains reservoir into a flooded area whether it is related to a low sensitivity of the conversion formulation or to the the volume of water considered for the





conversion and (3) to the lack of lateral expansion of the floodplains into the grid cells adjacent to the main river which, although it is partly covered by the overflow from the floodplains, may be underestimated due to some missing processes such as the groundwater lateral flow.

Concerning the first point, the satellite estimate of the flooded area may erroneously consider saturated soils as water surface (Zhou et al., 2021; Aires et al., 2018). Therefore the satellite based estimates of floodplain extent could cover open water surfaces, surfaces with a high soil moisture content and flooded vegetation. The model on the other hand considers separately soil moisture and open-water surface in the floodplains and rendering the floodplain extent difficult to compare. In GIEMS-2, the floods in the eastern region of the Pantanal appear to be more important during the wet season (DJF) compared to the flood
season (MAM) while other studies show that this regions is not regularly flooded (Padovani, 2010). There is a high correlation between GIEMS-2 flooded fraction and the soil moisture down to 0.5m depth from the surface in the NOFP simulation (cf. Supplementary Figure B7) which confirms the hypothesis of a saturated soil moisture signal in the GIEMS-2 dataset related to the precipitation during the rainy season. This saturated soil moisture is directly handled in the model by the soil hydrology and does not appear as a flooded area.

Finally, the major cause of underestimation of the flooded area is related to the limited lateral expansion of the floodplains. The floodplains scheme only considers the overbank flow and this limitation shows that some processes that need to be integrated in the model. For instance, the exchange between surface water and groundwater can raise the water table, therefore, driving large scale groundwater transports which are not foreseen by ORCHIDEE's routing scheme at a scale larger than the HTU (slow and fast reservoirs) and its corresponding grid cell. These processes are particularly important in the complex
hydrology of the Pantanal region (Junk and Wantzen, 2004; Freitas et al., 2019).

An implicit lateral transfer of moisture is carried out within the vadose zone through the soil moisture scheme (De Rosnay et al., 2000; de Rosnay et al., 2002; Campoy et al., 2013). As water in the floodplain infiltrates it will affect the soil moisture of the entire atmospheric grid cell and, thus, modify its exchanges with the atmosphere. This effect will be enhanced at lower resolution because the grid cells have larger areas and, therefore, the increase of soil moisture related to the floodplains
infiltration will affect more important areas. Therefore, there is a numerical diffusion at the resolution of the atmospheric grid which helps but has no physical cause. The soil hydrology is managed with a 1-D vertical model and, therefore, does not integrate the possibility to transfer laterally the increased soil moisture of the floodplains to the neighbour grid cells.

## 5   Impact of the floodplains on ORCHIDEE

### 5.1   Soil Moisture

The presence of the floodplains induces an additional infiltration into the soil. Figure 6 shows the mean soil moisture for the FP simulations and the relative difference with the NOFP simulations averaged between 1990 and 2013 considering: the period of maximum flooding (March, April, May), the dry period (September, October, November) and the entire year. The soil moisture is considered down to 0.5 meters below the surface because our main interest are the upper layers of the soil which are the most affected by the floodplains infiltration.





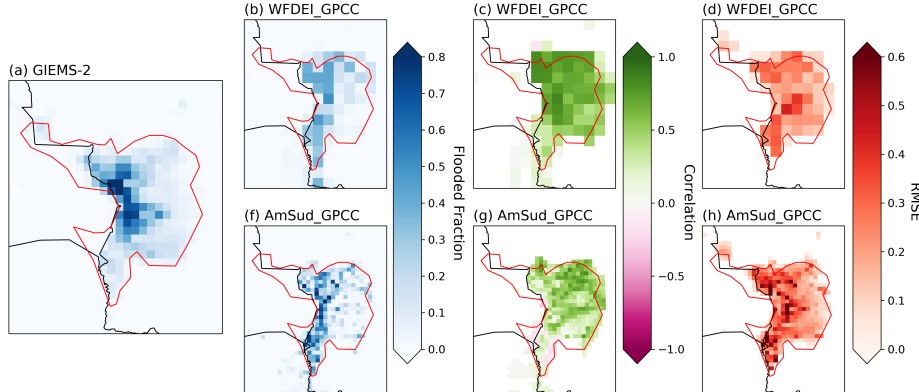

**Figure 5.** Mean flooded fraction in GIEMS-2 (a), WFDEI_GPCC (b) and AmSud_GPCC (f). Evaluation of the spatial representation of the floodplains in ORCHIDEE using the correlation (c and g) and the Root Mean Square Error (d and h) for WFDEI_GPCC (c and d) and AmSud_GPCC (g and h) compared to GIEMS-2.

The soil moisture increases over the most flooded area and it reflects the structure of the hydrological network in the Pantanal. The comparison with the NOFP simulation shows that these changes occur over a larger area during the flooded season compared to the rest of the year. The relative differences between the FP and the NOFP simulations reaches the highest values during the dry season because of the larger contrast with the dry conditions when no floodplains are considered.

The impact of floodplains on soil moisture is more important at lower resolution forcing because of the implicit numerical

diffusion occurring at the level of the atmospheric grid. This compensates the missing processes related to the shallow aquifers in the model. The groundwater flows and other missing riparian processes are quite complex but they can have an important impact on the soil moisture conditions (Krause et al., 2007; Frappart et al., 2011; Girard et al., 2003).

The introduction of the floodplains has also lead to the reduction of soil moisture on some grid cells close to the largest rivers in the region. The soil moisture decreases over some grid cells which are near the large rivers in the AmSud_GPCC_FP

simulations compared to AmSud_GPCC_NOFP. In this case, soil moisture receives water from the floodplains through infiltration while the floodplains collects part of the precipitation that would have gone directly into soil moisture elsewhere. If the infiltration from the floodplains supplying soil moisture does not compensate for the decrease of direct precipitation received, the soil moisture may decrease. This occurs in the North and Central East Pantanal over grid points with a low volume of water in the floodplains reservoir because there are no important rivers flowing through the grid cell. In this case, the infiltration

from the floodplains can be smaller than the amount of precipitation going into the floodplains reservoir instead of directly increasing soil moisture. This phenomenon is enhanced in regions with low infiltration rates (lower infiltration coefficient; cf. $k_{litt}$).



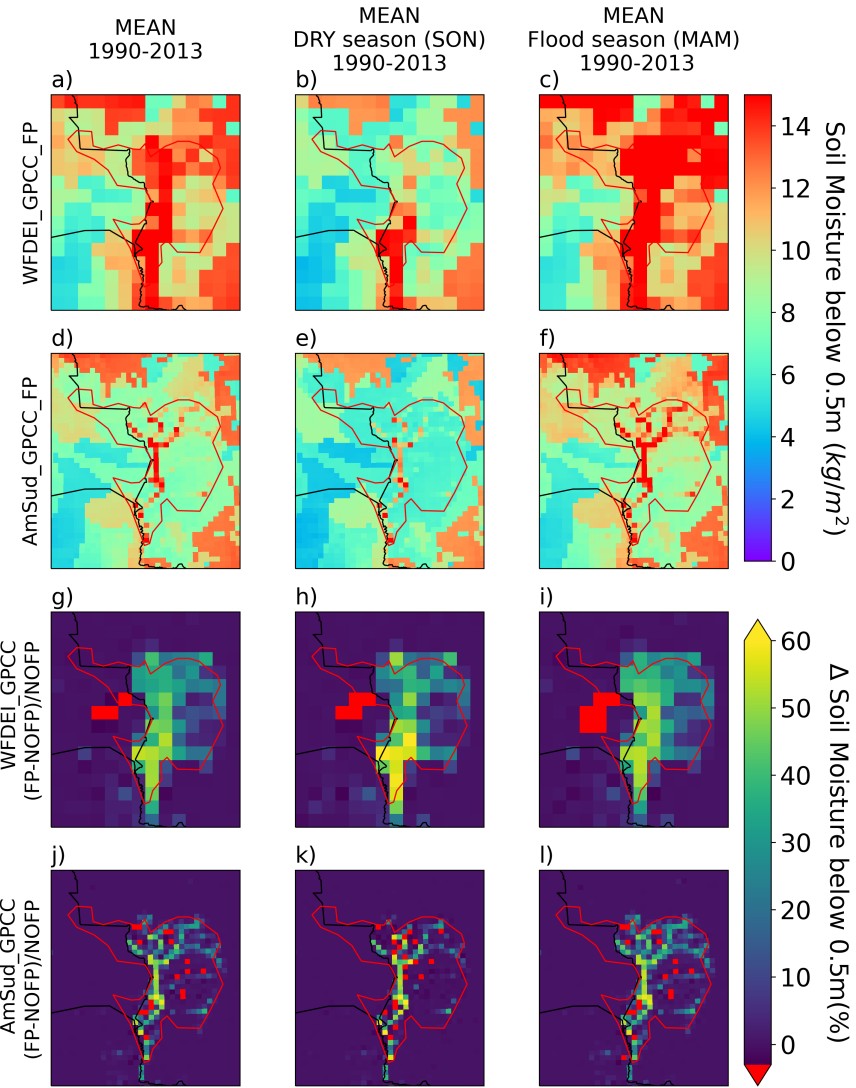

**Figure 6.** Mean soil moisture over the upper level (down to 0.5 meters below the surface) during the 1990-2013 period considering the full year (a,d,g,j), the dry season - SON (b,e,h,k) and the flood season - MAM (c,f,i,l) for the WFDEI_GPCC_FP simulation (a,b,c) and the AmSud_GPCC_FP (d,e,f). Difference between the FP and NOFP simulations for WFDEI_GPCC (g,h,i) and AmSud_GPCC (j,k,l).



## 5.2 Vegetation

The state of the vegetation is presented through the leaf area (LAI) variable which determines within ORCHIDEE also the
fraction of vegetation in the grid cell. Soil moisture, through plant transpiration and carbon assimilation, is one of the main
drivers of the vegetation and its LAI. This is why vegetation is also affected by the floodplains scheme.

Figure 7 shows, for each vegetation type existing in the Pantanal in ORCHIDEE, the ratio of the vegetation cover to the
maximum surface they can occupy during the flood season (no hatch) and the dry season (hatched) for the FP and NOFP
simulation, respectively in blue and orange for the WFDEI_GPCC (Figure 7.a) and the AmSud_GPCC simulations (Figure 7.b).
It should be noted that both simulations have the same vegetation description input. Although the LAI drives this process, we
consider here the vegetfrac/maxvegetfrac ratio as it allows to evaluate the development of the vegetation relatives to the
maximal vegetation cover it reaches for $LAI > 1m^2/m^2$.

All vegetation types are affected by the floodplains except for C3 crops in AmSud_GPCC. For most of these PFTs, the
difference only occurs during the dry season such as for natural C3 and C4 grasses and C4 crops. For the Tropical Broadleaf
(evergreen and raingreen) and for the temperate Needleleaf Evergreen, this change occurs during both the flooded and dry
seasons.. For short vegetation the floodplains mainly enhances vegetation fraction during the dry seasons as they have shallow
roots and thus only have access to upper soil moisture. For tall vegetation on the other hand the increased vegetation fraction
is more persistent as roots can exploit the increased deep soil moisture.

Some regions of the Pantanal see their vegetation fraction decrease with the activation of the floodplain scheme. This can be
explained by the reduction of the soil moisture related to the floodplains explained previously and observed in Figure 6.

The ratios of surface occupied by the PFT to its maximum are generally higher in the simulations forced by WFDEI_GPCC
which is related to the larger increase in soil moisture (cf. Figure 6). The changes between the FP and NOFP simulations are also
higher for WFDEI_GPCC compared to AmSud_GPCC which is related to the larger impact of the floodplains on soil moisture
in WFDEI_GPCC. These results shows that ORCHIDEE without floodplains is unable to develop the vegetation detected by
ESA-CCI and thus has a systematic bias in this region. Only activating the floodplain allows to develop the vegetation which
is observed. Therefore, the floodplains scheme is important for a more realistic simulation of the vegetation over these regions.
This occurs, for example, to the tropical broadleaf raingreen which are particularly affected by the floodplains scheme. This
vegetation type has an important presence in the North of the Pantanal (cf. Supplementary Figure B1). Without floodplains,
this vegetation type does not have enough soil moisture to grow correctly and cover the observed maximum area in the model.

The higher development of the vegetation in FP (driven by higher LAI values) also increases the roughness height for mo-
mentum and heat in the ORCHIDEE model (not shown). These variables have an impact on the turbulent exchange coefficients
in the calculation of latent and sensible heat within ORCHIDEE. Once coupled this impact will propagate to the planetary
boundary layer and also affect the momentum transport in the lower atmosphere.



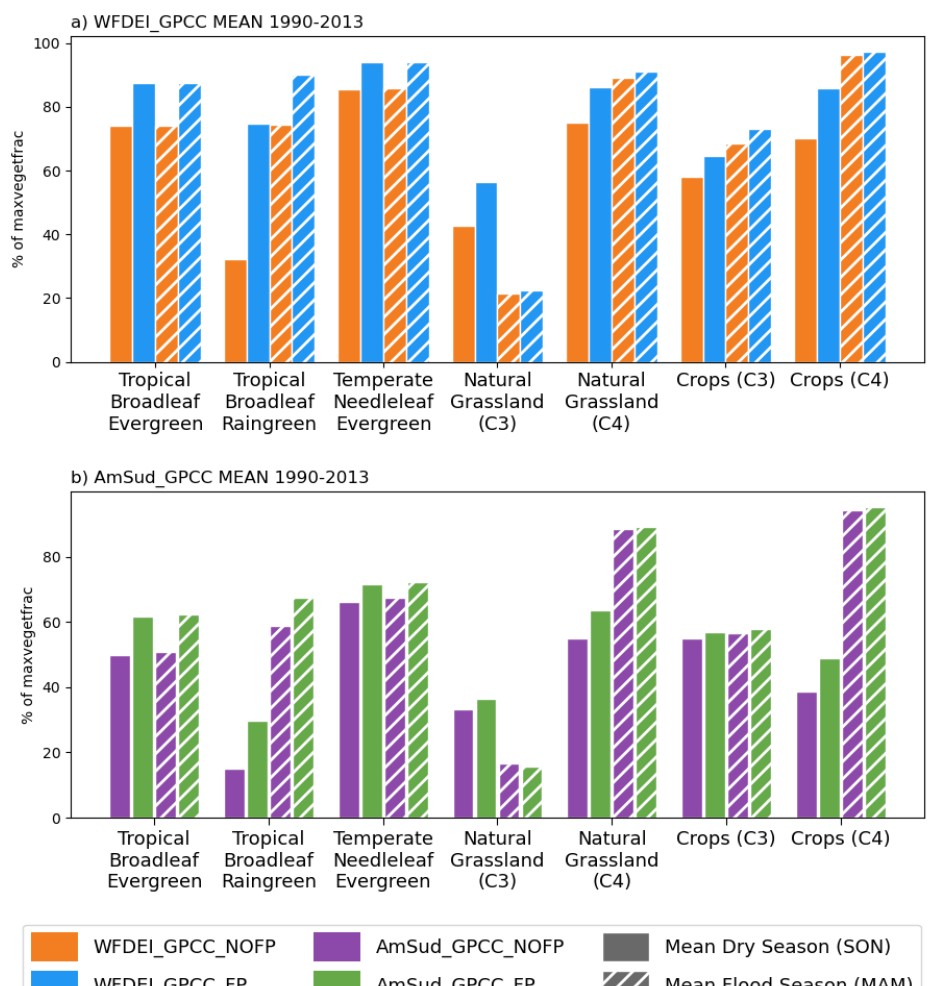

**Figure 7.** Bar plot of the percentage of the maximum vegetation cover in the model for the FP (blue) and NOFP (orange) simulations during the dry season - SON (no hatch) and during the flood season - MAM (hatched) for the simulations forced by WFDEI_GPCC (a) and AmSud_GPCC (b).

## 5.3 Surface Energy Budget

As seen in 3.2, over a large period of time (such as decades) net radiation can be partitioned between the latent and sensible heat flux. To compare the relative distribution of energy in latent and sensible heat flux, the evaporative fraction is shown in Figure 8 for the NOFP simulations and the FP simulations.





The evaporative fraction increases throughout the Pantanal region which means that the surface energy budget shifts to-wards energy lost to water phase changes. The latent heat fluxes increase while the sensible heat fluxes decrease. The largest
differences follow the spatial structure of the flooded area.

In both simulations, the highest values of mean evaporative fraction exceed 1 which is accentuated in the AmSud_GPCC_FP simulation. This means that over the main floodplains, the sensible heat fluxes becomes negative for some grid cells which indicates a surface cooler than the atmosphere over large period of time - i.e. on a 24 years average in this case (cf. equation 22). This behaviour is probably unrealistic in a tropical region and is related to the absence of feedback with the atmosphere. In
this off-line set-up of ORCHIDEE evaporative demand is high because air temperature and humidity have been established (In the re-analysis for WFDEI and the atmospheric model for AmSud_GPCC) without considering the floodplains. The lower atmospheric conditions thus become incoherent with the simulated surface conditions when the floodplains are activated. This also explains the higher values in AmSud_GPCC_FP compared to WFDEI_GPCC_FP due to the higher evaporative demand in this forcing. It shows that the changes brought about by floodplains are so fundamental in the surface energy balance that
they cannot be considered without the coupling to the atmosphere.

Figure 9 evaluates the averaged terms of the surface heat budget. The Ground Heat fluxes are not shown as they are negligible compared to the other fluxes. In equation 21, the sensible and latent heat flux are not the only variables which can be changed by the floodplains. Net surface radiation is also impacted by albedo changes induced by the stronger vegetation development when floodplains are activated.

In general, albedo decreases when the soil moisture increases and when the vegetation density increases. However, albedo may also vary depending on vegetation type and on soil types (Clay, Sand, Silt). depending on vegetation and soil types (clay, sand, silt). The vegetation cover increase will impact the albedo differently for different vegetation types. The vegetation type can have a higher albedo than the local bare soil, therefore, the albedo may increase when this vegetation type develops over regions with scarce vegetation, replacing a low albedo bare soil cover. The net shortwave radiation ($SW_{net}$) in the NOFP
simulations has values close to the FP simulations. It is, as expected by the analysis of the forcings, higher in AmSud_GPCC. This means that although the albedo slightly changes with the floodplains scheme, it doesn't have an important impact on the surface energy budget. The net longwave radiation ($LW_{net}$) slightly increases in the FP simulation compared to the NOFP induced by the decreased surface temperature.

The latent (Qle) and sensible (Qh) fluxes have more important changes between the NOFP and the FP simulations. The latent
heat flux increases in the FP simulation compared to the NOFP by 30% (WFDEI_GPCC) and 60% (AmSud_GPCC) while the sensible heat flux decreases by 70% in both forcings.

Therefore, the changes in surface temperature and energy balance find their principal origin in the impact of the floodplains on the latent and sensible fluxes instead of the changes in net radiation.

## 5.4 Evapotranspiration

The changes in the surface energy budget are dominated by modifications of the water available for evapotranspiration and enhanced by the development of vegetation. These impacts are reflected in the annual cycle of the different components





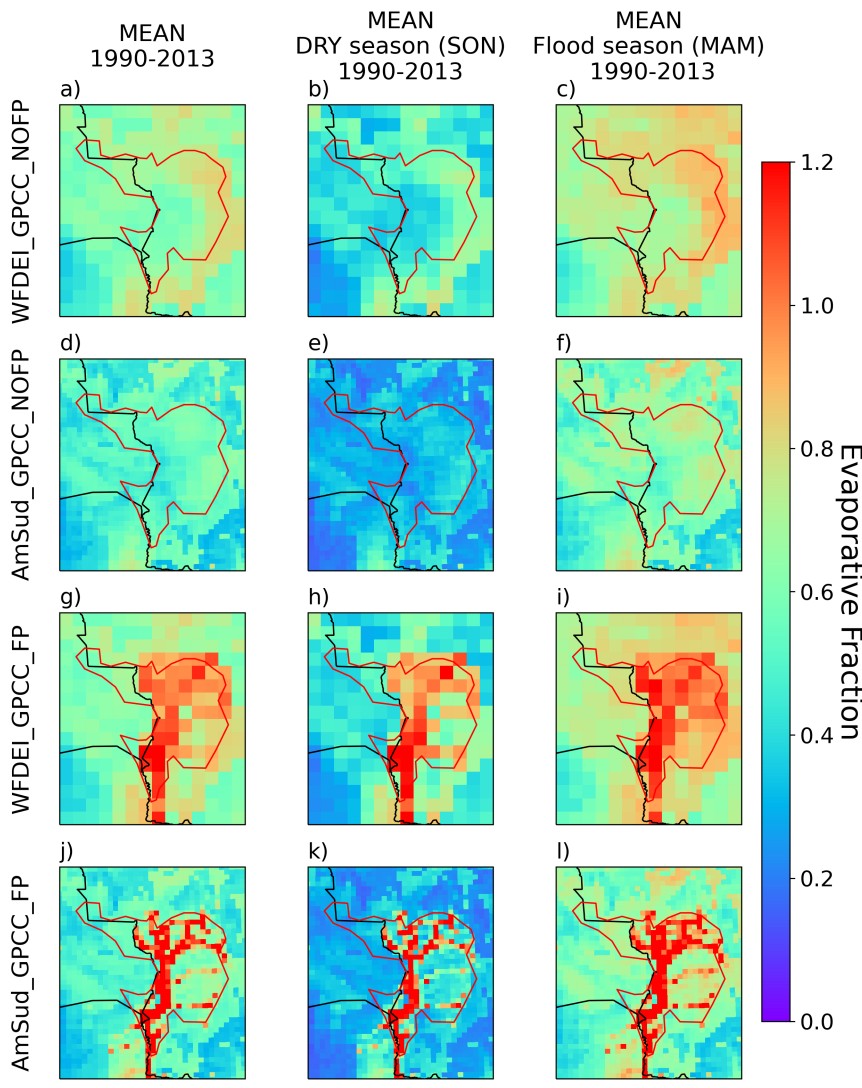

**Figure 8.** Mean Evaporative fraction for each grid cell in WFDEI_GPCC_NOFP (a,b,c) and AmSud_GPCC_NOFP (d,e,f), the WFDEI_GPCC_FP (g,h,i) and the AmSud_GPCC_FP (j,k,l) for the period 1990-2013 considering the full year (a,d,g,j), the dry season - SON (b,e,h,k) and the flood season - MAM (c,f,i,l).

of evaporation (bare soil, transpiration, floodplain evaporation and interception loss) which are shown for both atmospheric forcings in Figure 10.

Potential evaporation is lower throughout the year in the FP simulations when compared to NOFP. This is a direct conse-

quence of the decrease of surface temperature over the floodplains which will decrease the saturated surface humidity and the



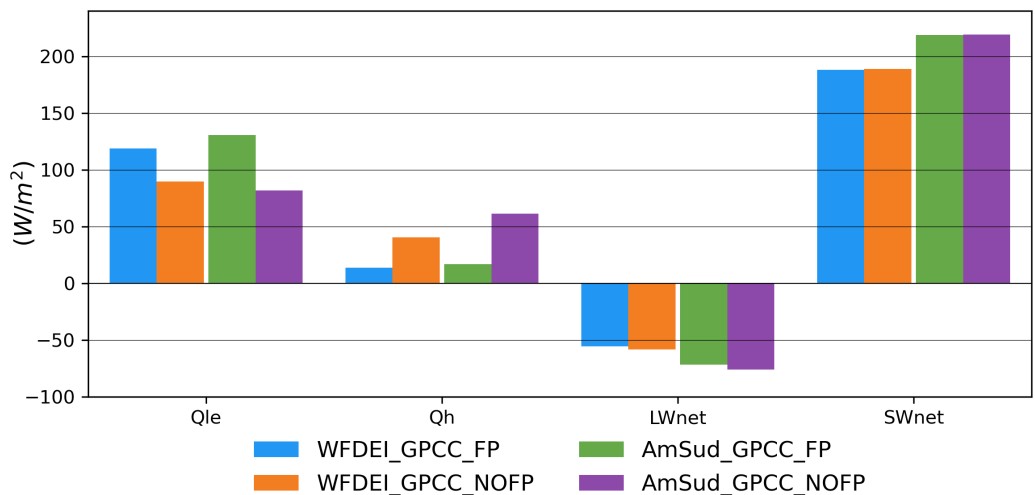

**Figure 9.** Average value over the Pantanal of the different terms of the equation 21 (Qle=LE, Qh=H) between 1990 and 2013.

lack of adjustment of the lower atmosphere which should be more humid than assumed in the forcing data sets. This is confirmed by the fact that the potential evaporation changes between NOFP and FP follow the same spatial structure as the surface temperature (not shown). Despite this decrease of the potential evaporation, the actual flux is higher in the FP simulations with the largest increases occurring between June and October which corresponds to the drier part of the year. This is a consequence

of the fact that evaporation over the area of the floodplains is water-limited when the lateral flows are not taken into account. It is only when water is allowed to converge in the floodplains that a sufficient amount of water becomes available to support evapotranspiration despite lower potential evaporation in FP. The floodplain evaporation brings the total flux to levels closer to those estimated in Schrapffer et al. (2020) through a water balance estimation using observations and numerical simulations. This leads us to believe that the evaporation is more realistic when the floodplains scheme is activated.

Bare soil evaporation is similar in both simulations although it is slightly higher in the NOFP simulation during the wet season. Bare soil evaporation is limited by the precipitation over bare soil and not affected by lateral transport of water. The amount of precipitation over bare soil is reduced in the simulation with floodplains because of the reduced bare soil area due to the increased vegetation fraction for most of the PFTs inducing a decrease of the rainfall over bare soil as the precipitation falling over the flooded fraction of the grid cells directly goes to the floodplains reservoir of the HTUs of the grid cell.

Transpiration is higher in the FP simulation between June and November. It is the largest change apart from direct open water evaporation. This change is explained by: (1) the increase of the LAI and of the vegetation fraction ($\mathrm{vegetfrac}$) in the FP simulation and (2) to the increased soil moisture available to support plant photosynthesis during the dryer part of the year. Comparing the transpiration between WFDEI_GPCC_FP and AmSud_GPCC_FP, there are higher values in WFDEI_GPCC_FP during the





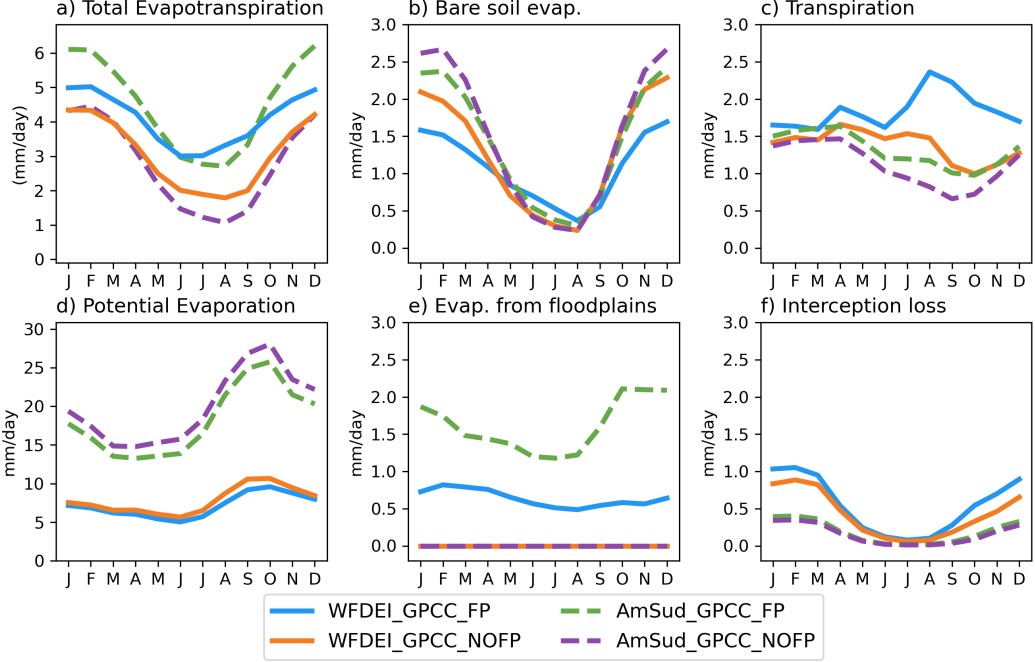

**Figure 10.** Annual cycle of the Evapotranspiration variables for WFDEI_GPCC (solid lines) and AmSud_GPCC (dashed lines) for the simulation with (blue) and without floodplains (red): (a) Total evapotranspiration, (b) Bare soil evaporation, (c) Transpiration, (d) Potential Evaporation, (e) Evaporation from floodplains and (f) Interception loss.

dry season. This difference is consistent with the impact on soil moisture described above and is another consequence of the observed numerical diffusion issue.

The interception loss is higher in the FP simulation compared to the NOFP simulation during the rainy season since the canopy in the FP simulation can intercept more water due to the higher LAI and the higher vegetation cover.

The other main difference is that the evaporation from the floodplains is much higher in AmSud_GPCC (around 1.5-2 mm/day) compared to WFDEI_GPCC (around 0.4-0.5 mm/day) which is related to the higher flooded area in AmSud_GPCC and to the higher incoming radiation in AmSud_GPCC when compared to WFDEI_GPCC.

The differences of evapotranspiration between FP and NOFP simulations have also been assessed spatially (not shown). They follow the spatial patterns of changes in evaporative fraction (cf. Figure 8). These changes in evaporative fraction are driven by both the increase of soil moisture and the presence of open water in the simulations with the floodplains scheme activated. In WFDEI_GPCC, these changes are mainly driven by the soil moisture changes while in AmSud_GPCC, they are dominated by the higher fraction of inundated area due to the numerical diffusion effect discussed above. Both resolutions are responding differently to the floodplains scheme which explains the need for different parameter values at each resolution.





Ideally, the floodplains scheme should behave the same way at all resolutions. However, the crude assumptions imposed by the definition of soil moisture at the atmospheric grid level and not the HTU scale explains most of the differences. It thus seems important in these regions of important horizontal surface moisture convergence to link soil moisture to the hydrological
rather than the atmospheric grid of a land surface model. The use of a specific sub-surface component such as suggested in the framework for LSM described in Hallouin et al. (2022) can be used to solve these issues. The resolution of the forcing has an impact on the relative importance of the different floodplains processes such as the balance between soil moisture and flooded area. However the impact of the floodplains on the evapotranspiration and on the land-atmosphere fluxes is similar at all resolutions. Moreover, it should be added that there is a lack of observations over the region which complicates the
development of the parameterization of this type of model.

## 5.5 Temperature

Surface temperature ($T_s$) determined by the surface energy balance at the surface and will thus be directly affected by the impact of floodplains on evaporation.

The Pantanal is in a tropical region which receives large radiation fluxes throughout the year. Therefore when floodplains are
not considered, the underestimation of water brought by the convergence of rivers in the floodplains leads to an underestimation of evaporative cooling and, therefore, an overestimated surface temperature.

Due to its dry bias, AmSud_GPCC atmospheric forcing has a higher near surface temperature and specific humidity than WFDEI_GPCC. This explains the higher surface temperature in AmSud_GPCC_NOFP compared to WFDEI_GPCC_NOFP.

The activation of the floodplains scheme reduces surface temperature over the Pantanal (cf. Supplementary Figure B4)
driven by the increase of evaporation described previously. The difference of temperature is lower for the WFDEI_GPCC forcing (around -1∘C) compared to AmSud_GPCC (with differences up to -3∘C during the flood season and up to -6∘C during the dry season).

Observational temperature dataset such as CRU TS4 (Harris et al., 2020) does not reflect any hydrological pattern of temperature difference over the Pantanal (cf. Supplementary Figure B5 and B6). This is due to the scarcity of in-situ observations
over the region and the coarse resolution of observational datasets which interpolate these in-situ observations.

The distribution of the average daily temperature over the most flooded parts of the Pantanal (with a mean $\text{flood\_frac} > 0.1$) in both forcings is shown in Figure 11. During the Flood Season (MAM, Figure 11.a), the activation of floodplains reduces both maximum and minimum values for AmSud_GPCC and WFDEI_GPCC. In both cases, the body of the distribution (the distribution between percentile 10 and 90) is shifted toward lower temperature and is more concentrated as can be seen by the
change in the shape of the distribution of values.

During the Dry Season (Figure 11.b), the body of distribution of the temperature as well as the extremes are shifted toward lower values. The reduction of minimum temperature is more important in AmSud_GPCC whose minimum is 3° lower than the minimum of AmSud_GPCC_NOFP in both seasons.

In conclusion, the increased evapotranspiration dampens temperature extremes caused by meteorological and radiation fluc-
tuations.





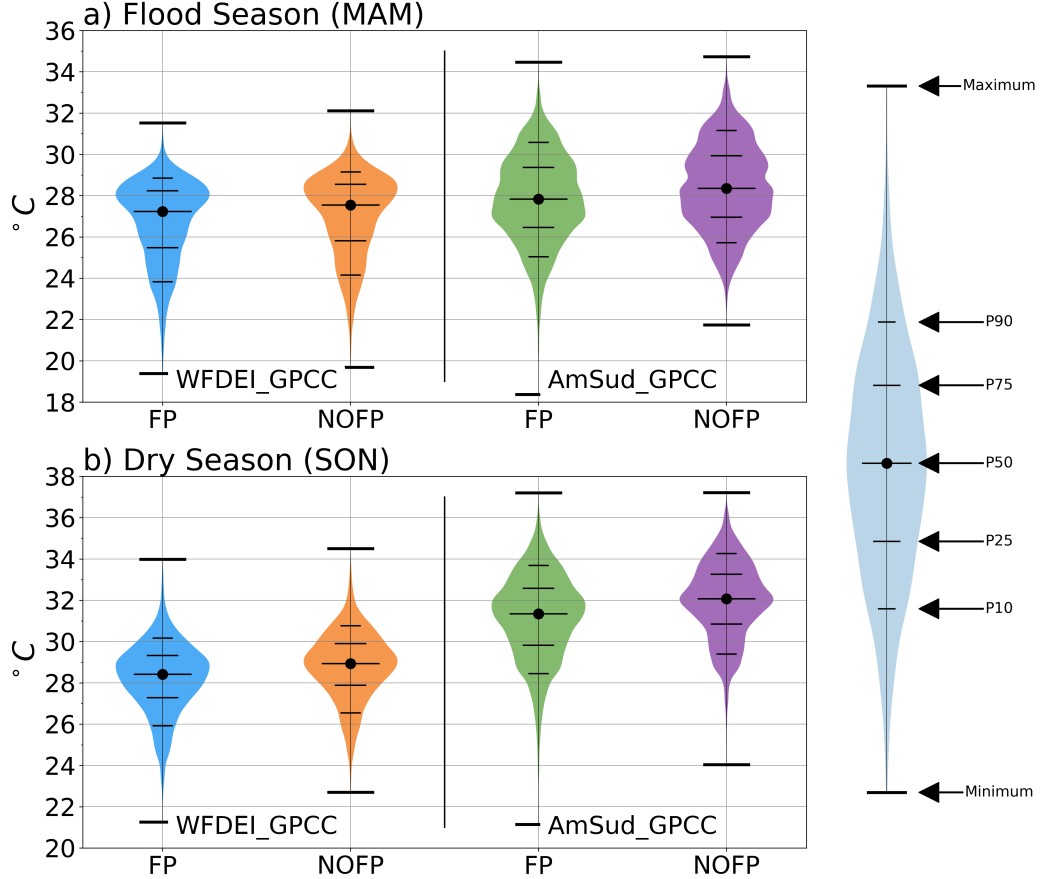

**Figure 11.** Representation of the distribution of the average daily temperature over the most flooded part of the Pantanal for the pair of simulation with (left) and without floodplains (right) forced by WFDEI_GPCC and AmSud_GPCC during the period 1990-2013. The extremum, the median as well as the percentile 10, 25, 75 and 90 are represented.

## 6 Discussion and Conclusion

With the progress made in the description of surface flows in land surface models and especially with increasing resolutions, the parameterization of the interactions of these flows with the landscape need to be revised. In this article, we proposed a methodology to implement the representation at higher resolution of the floodplains in a land surface model and, as a first

benchmark, evaluate its performance over the *Pantanal* region in South America performing offline simulations (forced by atmospheric forcings) at different resolutions. Figure 12 summarizes the impacts of the presence of floodplains on the land surface variables in ORCHIDEE.



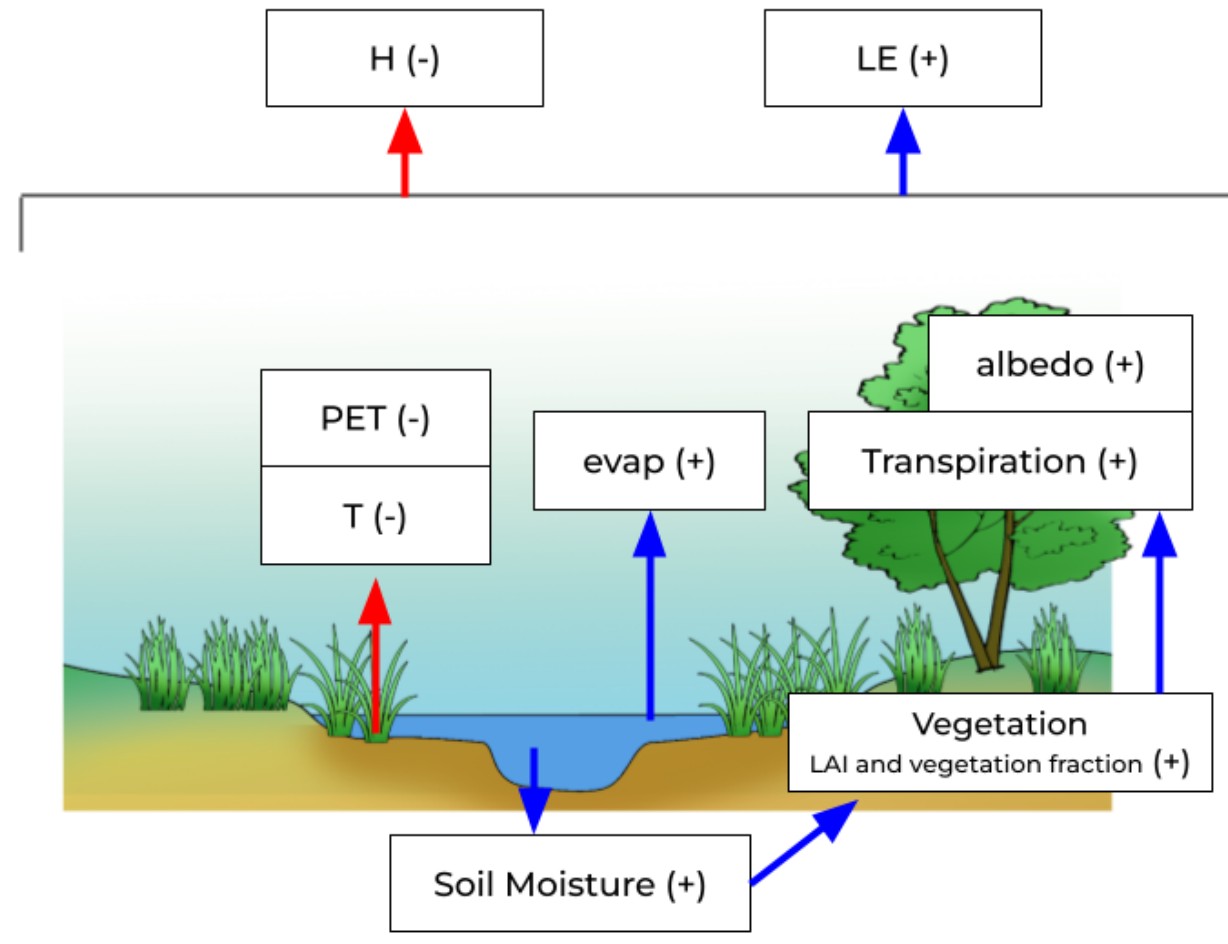

**Figure 12.** Summary of the different impact of the floodplains over the land surface variables in a Land Surface Model.

Offline simulations were performed to validate and analyse the functioning of the floodplain scheme. A pair of experiments for each atmospheric forcing has been performed, one with (FP) and another one without floodplains (NOFP), and allows to explore the role of atmospheric forcing uncertainty and resolution on our ability to reproduce the impact of floodplains on surface/atmosphere exchanges. The pair of simulations has been forced by the atmospheric conditions with different resolutions and different atmospheric water demand but same precipitation. These forcings underestimate the near-surface humidity and temperature because they didn't consider the impact of the floodplains on the atmosphere and because these regions have scarce in-situ observations, however, the 20 km forcing has a higher atmospheric evaporative demand due to its lower near surface humidity and higher near surface air temperature and incoming radiation.

The floodplains scheme presented in this article exploits the high resolution information of the hydrologically coherent Digital Elevation Model used to construct the HTUs. This allows us to describe the shape of the floodplains with more precision.



The resolution of the hydrological units increases compared to the previous version of the floodplains scheme (D'Orgeval, 2006). The exploitation of the high resolution river graph allows to improve the description of the floodplains within the hydrological units and to parameterize the overbank flow of the HTUs. The infiltration of the floodplains into the soil is a crucial aspect as it permits the floodplains to affect a larger area. However, the soil moisture in ORCHIDEE is still managed on the atmospheric grid level and, therefore, creates an uncontrolled numerical diffusion which will have to be addressed in the future. This new version of the floodplains scheme integrates the possibility for the water in the floodplains to overflow in the floodplains of the upstream HTUs which was not possible in the previous version and is crucial at higher resolution. The calibration of the parameters could be avoid if we can define them based on physical relationship. However, this is not always possible because the calibration of these parameters can hide some missing processed such as it is the case with the floodplains infiltration adjusted as the complex vegetation and soil of the floodplains (flooded vegetation and soil covered by sediments).

The representation of the water cycle over the Pantanal has been assessed by comparing the discharge simulated at the station of *Porto Murtinho* which is at the outflow of the Pantanal. In both cases, the activation of the floodplains scheme improves the simulation of the discharge at the outflow of the Pantanal by shifting the peak flows and reducing its amplitude. Still, the intra-annual variability of discharge is overestimated for both forcings but less so in the higher resolution version. The water mass in the floodplains has only a limited impact on the total water storage and, thus, could not be validated with GRACE data. Although the floodplains represent a relatively small volume of water, they have an important impact on river discharge and on surface atmosphere interactions.

The mean flooded area is coherent between both AmSud_GPCC_FP and WFDEI_GPCC_FP simulations and the annual cycle is underestimated for both forcing compared to different satellite estimates. However, there are epistemological difficulties in defining "flooded area" and large discrepancies between the various satellite estimates considered here. It is difficult to correctly assess the flooded area (Schrapffer et al., 2022, in press) principally in areas covered by floods and over saturated soils and, therefore, to correctly identify the deficiencies of the model. The representation of the flooded area is a major issue for any flooding scheme. Under the assumption that the precipitation from the GPCC dataset is correct over the Pantanal and knowing that the river discharge is close to the observation when the floodplains scheme is activated, it can be concluded that the water cycle of the catchment is correctly represented in the model. Therefore, the issues can come from: (1) the fact that there exists different ways to define what a flooded area is (Schrapffer et al., 2022, in press), (2) the conversion of the volume of water in the floodplains to flooded area, (3) missing processes such a groundwater transfers. The estimate of flooded area can take different forms such as the presence of shallow water or flooded vegetation and can be related to some groundwater resurgence. The satellite products consider the open-water surfaces and the regions with high soil moisture content while they are considered separately in the model. In the model, the flooded area is modeled through the relationships between the volume / height of the floodplains and the flooded area. More complex methods could be considered if elevation would be known to a higher accuracy in these flat areas. Yet, it should be remembered that there are still large uncertainties in the DEM over flat regions and even more if there is an important vegetation cover (Yamazaki et al., 2019). Several tests have been performed to evaluate the sensitivity of the predicted flooded area from the water volume in the floodplain reservoir. These tests have shown that this volume/area function doesn't seem to play an important role at large scale on estimated total flooded area over the





region. ORCHIDEE uses a convergent flow model however, in some cases, the floodplains can be caused by a divergent flow such as the *Taquari Megafan* over the *Taquari* river in the central region of the Pantanal. The representation of this type of process would require a high precision hydrological DEM as the floodplains in this region are very sensitive to small differences in orography. The divergent processes are not represented in the Hydrological DEM and, therefore, are not implemented in ORCHIDEE. The groundwater fluxes between grid cells are not considered either in the actual versions of ORCHIDEE. The water that infiltrates from the floodplains reservoir of an HTU affects only local soil moisture and is not transported in the saturated layers of the ground. However, this increase of the soil moisture should be able to affect the neighbouring grid cells (Krause et al., 2007; Frappart et al., 2011; Girard et al., 2003). These small scale processes are implicitly integerated for coarse atmopsheric resolutions, such as the 0.5° grid used here. However, at higher resolution, these processes need to be represented explicitly. The soil moisture would be better represented if it was calculated at the resolution of the river graph, i.e. at the HTU level. However, this will aggravate the lack of the uncontrolled numerical diffusion currently occurring and thus call for a physical representation of the horizontal diffusion in the saturated soil layers. To do so, the hydro-geological structures within the Pantanal would need to be informed and used in the representation of the horizontal diffusion.

The vegetation dynamic is strongly affected by the flooding and the soil moisture increase. The infiltration of water from the floodplains into the soil increases the vegetation density even during the dry season. This impacts on ORCHIDEE's other surface properties such as the roughness height for momentum and the roughness height for heat. The albedo decreases over almost all the Pantanal but increases over the most flooded part due to the covering of the dark soil by a vegetation type with a higher albedo. The maximal potential vegetation cover in ORCHIDEE for a vegetation type is constructed from satellite derived products but these vegetation types require a realistic water availability to grow to their full potential in the model. Without floodplains, some PFTs derived from satellites do not thrive. The improved coherence between the potential vegetation cover and the development of the vegetation in the model is a good indicator of the fact that the floodplains scheme improves the land surface simulations over the Pantanal.

Land-atmosphere fluxes over the Pantanal are also affected by the activation of the floodplains scheme with the sensible heat fluxes diminishing while the latent heat fluxes are increasing. This is coherent with the surface temperature decrease over the floodplains. The changes between the FP and NOFP simulations showed that the net radiation change is low compared to the sensible and latent heat fluxes, thus, the balance between the two fluxes are the main changes in the surface energy budget.

The model allows us to observe the different origins of the changes in the latent heat fluxes. The principal contribution to the increase of evapotranspiration is transpiration followed by open water evaporation. The transpiration also plays an important role, a consequence of the increase of soil moisture and vegetation density. However, its impact is more limited over the Pantanal at higher resolution due to the ill controlled numerical diffusion of soil moisture and lacking horizontal moisture diffusion in the saturated layers of the soil. The potential evaporation is lower in the FP simulation due to the decrease of surface temperature but does not affect the actual flux as the increased water availability dominates.

The absence of coupling leads to high and unrealistic values of evaporative fraction showing that it is possible that the latent heat fluxes may be overestimated and the evapotranspiration too. This has been observed more clearly in the forcing AmSud_GPCC which represents an atmosphere with a higher evaporative demand compared to WFDEI_GPCC. The Am-



Sud_GPCC and WFDEI_GPCC are issued from AmSud and ERA-Interim which both do not include floodplains. The near surface observations available in the regions and used to bias correct the re-analysis are insufficient to compensate from this lack in the model. The forcings don't integrate the impact of the floodplains on near atmosphere conditions and are not reacting to the surface conditions. Thus, despite the higher evapotranspiration the near surface humidity remains low and enhances, more than it should be, the evapotranspiration. The coupling between ORCHIDEE and an atmospheric model will help analyse the impact of the floodplains under more realistic conditions (with feedback from the atmosphere) and give the opportunity to better analyze the land-atmosphere interactions.

Over the Pantanal, the floodplains scheme seems to capture the dominant hydrological processes involved but, looking at the subject with a wider angle, other types of processes could be critical in regions we refer to as floodplains. For example, the open water surfaces can also be related to some ponds or lakes, to different types of flooded forest or to swamps. The configuration proposed here is not optimal for these other types of wetlands. Some other schemes can be constructed from the spatial description of these processes and their interaction with the atmosphere such as for ponds, which are small lakes flooded due to the local precipitation, or swamps and flooded forest that can be managed by redirecting small fractions of river flow into soil moisture in areas of long residence time. IMaps are needed to discriminate between all the different types of flooded area which have different dynamics and must be parameterized differently from the floodplains scheme descrie in this paper.

To conclude, the different impacts of the floodplains scheme are coherent to what can be expected by the presence of open water and enhanced infiltration over floodplains. Although the evapotranspiration is overestimated compared to values derived from the model-based water balance in Schrapffer et al. (2020), the simulation with floodplains brings a more realistic representation of the land surface over the Pantanal region.

The developments illustrated in this paper show the differential in resolution and complexity between the representation of surface water and soil moisture or groundwater processes. This calls for an effort to refine the representation of water in the soils so that the gap in complexity and refined of processes is closed and the scheme is less reliant on ill controlled numerical artefacts.

The impacts of the increased resolution of the floodplains scheme in the ORCHIDEE model over the different surface variables have been better understood. These variables are expected to have an effect on the atmospheric boundary layer which has in turn an impact on the regional circulation and precipitation. To evaluate these feedbacks a coupled simulation with and without floodplains has been carried out and will be subject of a future publication. It is also crucial to assess the generalization of the modeling and calibration of the floodplains scheme over other large tropical floodplains such as the Inner Niger Delta, the Congo, the Amazon or the Sudd. This is crucial to advance our understanding of land atmosphere interactions over these important ecosystems.

*Code and data availability.* The code version and data used for this study are available at https://zenodo.org/record/7761859#.ZBuDFNLMLBc. It contains the ORCHIDEE code used for the simulations, the parameterization used in each simulation and the routing file used for the simulations. The code of the preprocessing tool to generate the routing files is available in Polcher et al. (2022) (https://doi.org/10.5281/



zenodo.7058895). The GRACE data has been extracted using Google Earth Engine. The Global Lake and Wetland Database is available from https://www.worldwildlife.org/pages/global-lakes-and-wetlands-database. The discharge used for Porto Murtinho is available from the Brazilian Agencia Nacional de Aguas (ANA, https://www.snirh.gov.br/hidroweb/).

### Appendix A: Statistical indexes

Statistical indexes are precious tools to quantify and discuss the performance of a model. For this reason, different statistical indexes are used in this article to evaluate the behaviour of the simulated discharge at the Porto Murtinho station compared to the observed values. The statistical indexes used in this article are described in the present section, the mathematical formulations of these indexes use the following nomenclature: $N$ is the total number of time steps considered, $M_t$ represent the model value at the timestep t and $O_t$ the observation corresponding to this time step, $\bar{O}$ will represent the mean value of the observations

over the interval of timesteps $[1, N]$.

The Nash-Sutcliffe model efficiency coefficient (NSE) allows us to compare the performance of the model to the mean value of the corresponding observed variable. It can be calculated by the equation (A1). Its values are in the range $]-\infty, 1]$ with 1 corresponding to a model perfectly representing the observed variable. For values of NSE lower than 0, the variable might be better estimated by the mean value of the observations.

$$NSE = 1 - \frac{\sum_{t=1}^{N}(M_t - O_t)^2}{\sum_{t=1}^{N}(M_t - \bar{O})^2} \tag{A1}$$

The Percent Bias (PBIAS) is an indicator allowing to evaluate systematic bias in the model compared to the observations. Positive values means that the model might be overestimating the variable while negative values mean the opposite. The equation of the PBIAS index is represented in the equation (A2).

$$PBIAS = 100\% * \frac{\sum_{t=1}^{N}(M_t - O_t)}{\sum_{t=1}^{N} O_t} \tag{A2}$$

The Root Mean Square Error (RMSE) is a classical index which is used to evaluate the performance of the model. The RMSE is a positive number representing the error in the same unit as the variable evaluated. It can be calculated by the equation (A3).

$$RMSE = \sqrt{\frac{\sum_{t=1}^{N}(M_t - O_t)^2}{T}} \tag{A3}$$

The correlation between the simulated and observed discharge is also presented along with the information of the significance of this correlation at a 95% level using a two-tails test.



# Appendix B:  Complementary figures

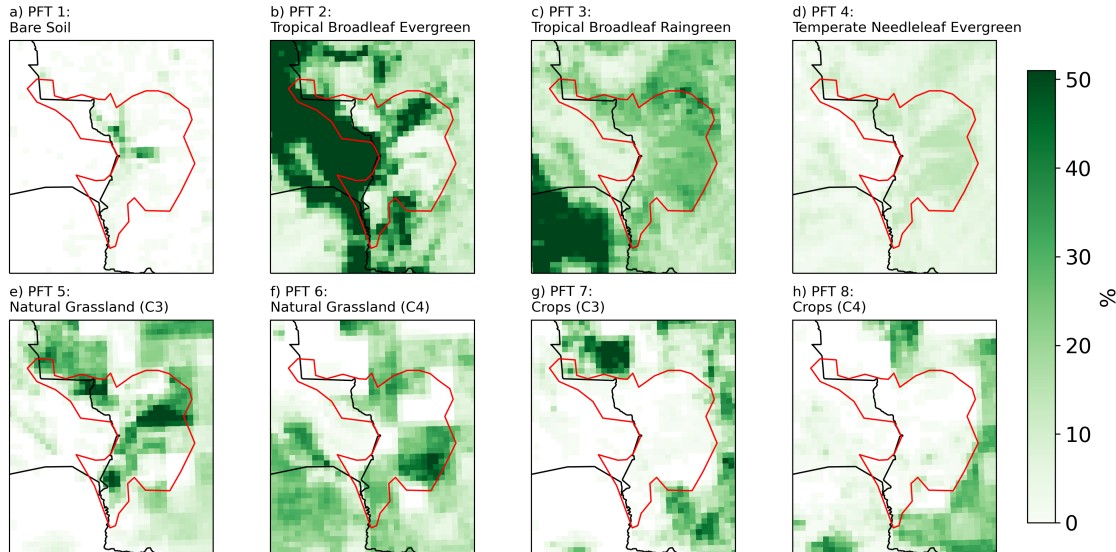

**Figure B1.** Description of the potential vegetation cover (maxvegetfrac) for all the vegetation types (PFT) existing over the Pantanal in the simulations. The PFT are constructed from the ESA-CCI database (European Space Agency-Climate Change Initiative; ESA, 2017).



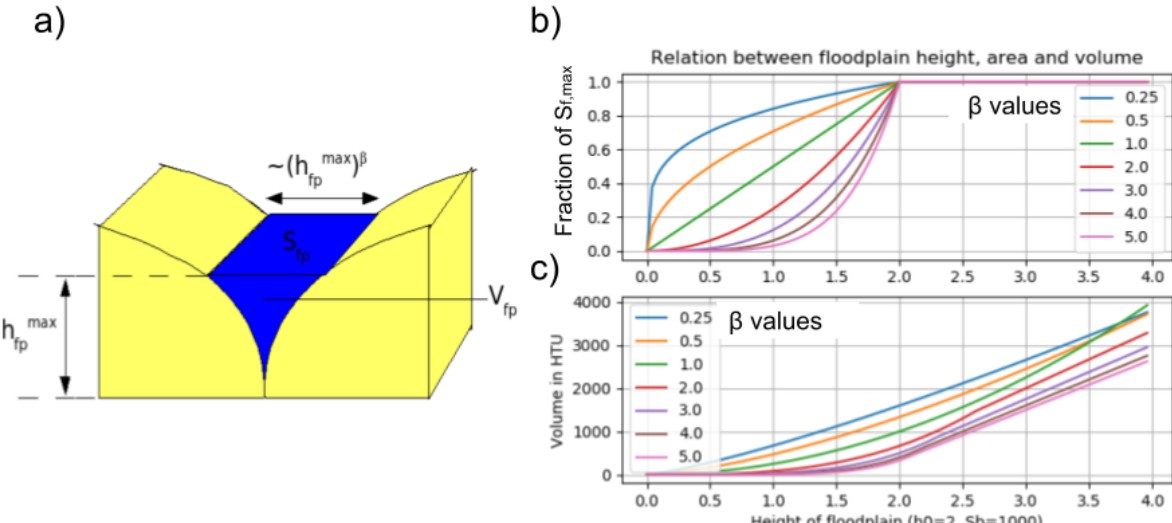

**Figure B2.** (a) Figure 4.9 from Tristan d'Orgeval's thesis (D'Orgeval, 2006) representing the parameterization of the floodplains shape, (b) relationship between the floodplains area and floodplains height depending on $\beta$ parameter with $h_0 = 2m$ and $f_{max} = 1$ and (c) relationship between the volume in the floodplains reservoir and the height of the floodplains depending on the value of the $\beta$ parameters for $h_0 = 2m$.



**Figure B3.** Annual cycle of the variables in the atmospheric forcings WFDEI_GPCC and AmSud_GPCC between 1990 and 2013 over the Upper Paraguay River Basin (UPRB).





**Figure B4.** Average Surface Temperature in the NOFP simulation forced by WFDEI_GPCC (a,b,c) and AmSud_GPCC (d,e,f) and the difference between the FP and NOFP simulation for WFDEI_GPCC (g,h,i) and AmSud_GPCC (j,k,l) between 1990-2013 period considering the full period (a,d,g,j), the dry season (b,e,h,k) and the flood season (c,f,i,l).



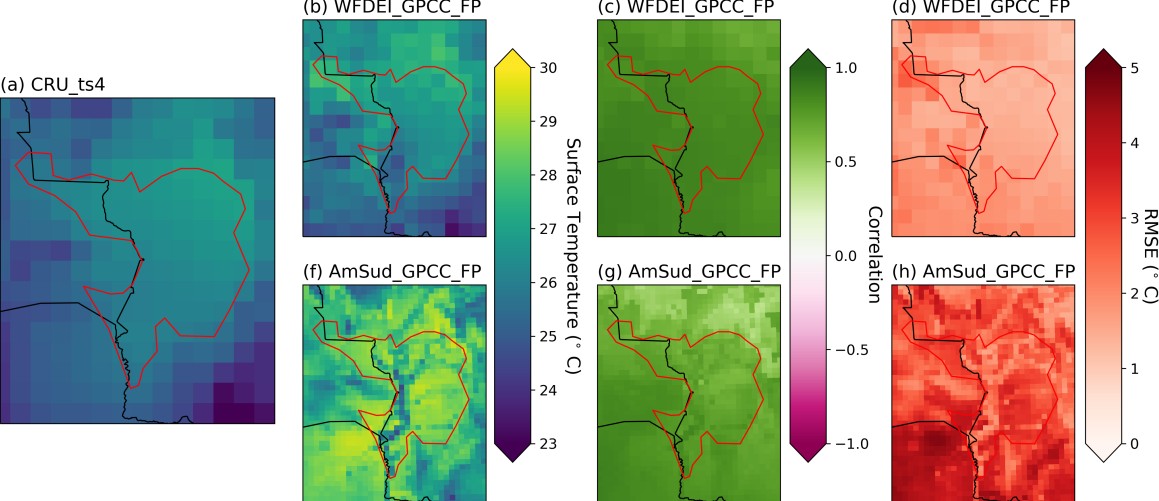

**Figure B5.** Mean surface temperature in CRU-TS4 (a), WFDEI_GPCC_FP (b) and AmSud_GPCC_FP (f) and comparison of the simulation with floodplains with CRU-TS4 using the correlation (c and g) and the Root Mean Square Error (d and h) for WFDEI_GPCC (c and d) and AmSud_GPCC (g and h).



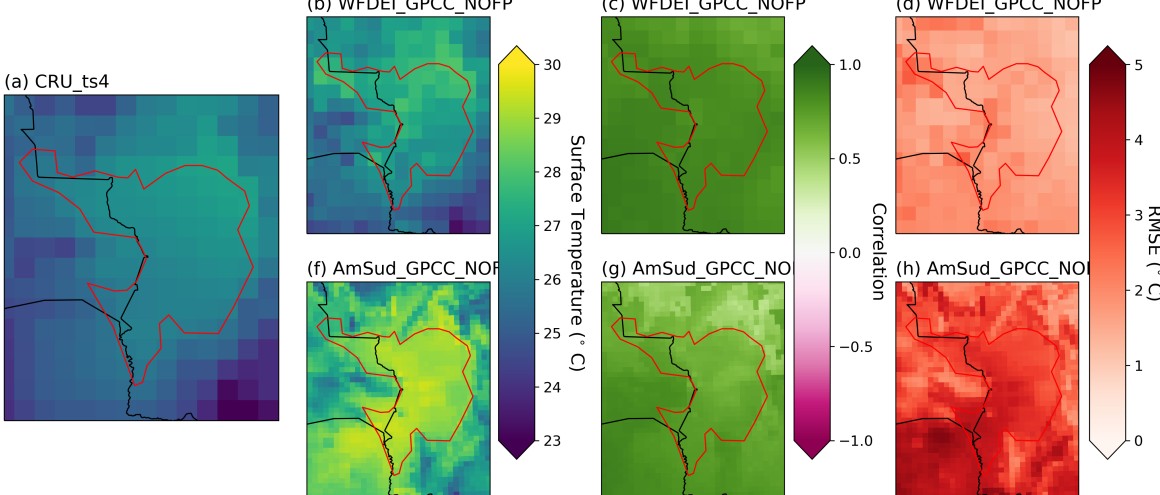

**Figure B6.** Mean surface temperature in CRU-TS4 (a), WFDEI_GPCC_NOFP (b) and AmSud_GPCC_NOFP (f) and comparison of the simulation without floodplains with CRU-TS4 using the correlation (c and g) and the Root Mean Square Error (d and h) for WFDEI_GPCC (c and d) and AmSud_GPCC (g and h).



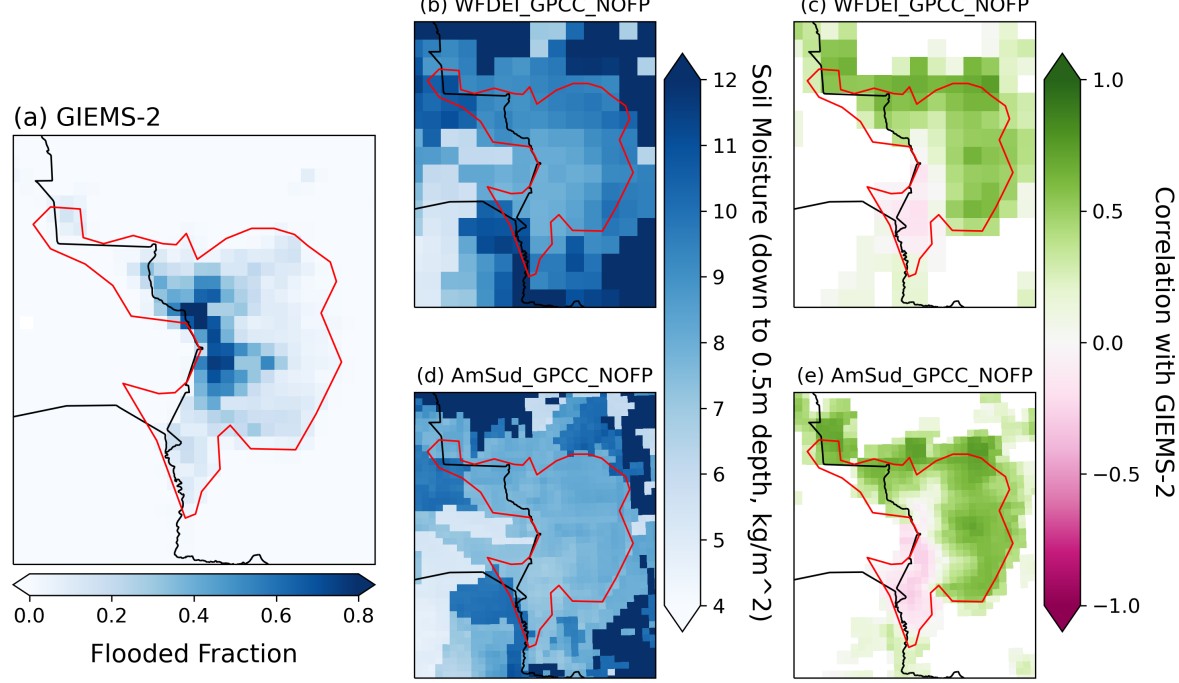

**Figure B7.** Considering the period 1992-2013, mean flooded fraction in GIEMS-2 (a), mean soil moisture down to 0.5m depth in WFDEI_GPCC_NOFP (b) and in AmSud_GPCC_NOFP (d), correlation between GIEMS-2 flooded fraction and the mean soil moisture down to 0.5m depth from the surface in WFDEI_GPCC_NOFP (c) and in AmSud_GPCC_NOFP (e).



*Author contributions.* Anthony Schrapffer designed the model, developed the code, designed and executed the numerical evaluations, Jan Polcher contributed to the model design, code development and evaluation, Anna Sörensson and Lluís Fita contributed to the model design and evaluation.

*Competing interests.* The authors declare that they have no conflict of interest.

*Acknowledgements.* We would like to gratefully acknowledge the support of the Agencia Nacional de Promoción Científica y Tecnológica (ANPCyT), Argentina (PICTs 2017-1406, 2018-02511); the Consejo Nacional de Investigaciones Científicas y Técnicas (CONICET), Argentina (PIP 11220200102141CO); the French-Argentina project ECOS Sud 2018 co-financed by the Ministerio de Ciencia, Tecnología e Innovación (MINCyT), Argentina and the Université Sorbonne Paris Nord, Francia (ECOS-A18D04); and the French National LEFE program (Les Enveloppes Fluides et l'Environnement; LEFE 12962). This work was granted access to the HPC resources of IDRIS under the
allocations 2019-A0070111113, 2019-A0080111527, 2020-A0090111113 and 2021-A0110111113 made by GENCI.

We would also like to acknowledge the Producers of GIEMS-2 (Prigent et al., 2020) for providing their satellite-derived surface water extent product which has been helpful to assess the flooded area in the model.



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
