# Peer review of "Introducing a new floodplain scheme in ORCHIDEE (version 7885): validation and evaluation over the Pantanal wetlands"

_EGUsphere, 2023_

## Referee Comment (RC2)

**General Comments**

This manuscript describes the new floodplain scheme implemented in ORCHDEE model, evaluate the validity of the new scheme, and analyzed its impact on other land surface variables. Even though it's still a case study simulation over Pantanal, I feel the paper very carefully analyzed how floodplain is important for land surface modelling.

The modeling strategy seems to be a bit complicated, while I feel the complexity is necessary given that the floodplain inundation itself is a complex physical process. I suggest the authors to provide more kind explanations about floodplain parameterization scheme, for example by using schematic figures, to help readers to understand how the proposed floodplain scheme works. However, the manuscript is overall well written, while minor revision is needed before acceptance.

**Major concerns**

[1] I feel the manuscript is too long. It might be unavoidable as a model description paper, but readability might increase if not-so-important parts are moved to supplements.

[2] So many variables/symbols are used to parameterize proposed floodplain scheme, and I feel difficulty following the explanations and equations. I suggest to create one schematic figure which represents the parameterization concept of floodplain scheme (with explicit description of which symbols corresponds to which variables). Visual explanation must help readers to understand about the new floodplain scheme.

**Specific comments:**

L193: whether the floodplains are activated or not.
This should be "regardless of whether …"
In addition, please explain what slow and fast reservoir represents. It is explained in results section that they represent aquifer and shallow groundwater, but this should be stated here. Otherwise, readers cannot know why they have limited relationship to floodplain scheme.

L235: The floodplains scheme allows a specific HTU to "overflow" the content of its floodplains reservoir into connected upstream HTUs with floodplains.
This is very interesting scheme. I wonder what is the impact of this overflow scheme on simulated water and energy budget. If space allows, please include some analysis.

L284 2.4.1: Case $S_{f,I} < S_{fmax,i}$

I recommend you to explain the case in plain language in the section title, not by the equation.

L285: height of the floodplain
This term is ambiguous. Do you mean "water surface elevation of the floodplain"?

L331: in order to define a mask of potentially flooded areas based on the following categories:
Could you please explain in which case this floodplain mask is required, and what is the impact of using this floodplain mask?

L355: before using the scheme over another region to evaluate if this parameterization is the more appropriate.
In many part of the world, there is no observation data for calibration. If possible, it's better to perform some sensitivity tests of parameters (confirm results are not so sensitive to parameters, or specify which parameter has larger impact).

L360: Methodology of Validation and Analysis
Please also provide some description of the simulation domain. Probably, a figure showing the simulation domain (with location of the gauges) is better to be provided.

L419: forced with ERA5 re-analysis data.
I assume this is regional atmospheric simulation, and in that case ERA5 must be "boundary condition" rather than "forcing".

Figure 2:
Could you please analyze the mechanics of river discharge delay? E.g. where water stays before reaching to the river gauge? Did they stay in floodplain as surface water? Or did they stay in soil by infiltration? Given that the difference between FP and NOFP simulation is large, it's better to provide detailed analysis on the mechanism which cause the difference.

L507: soil moisture and in the stream reservoir increases slightly
Considering the magnitude of change, compared to other storage variables, I feel the soil moisture was "significantly" increased by floodplain scheme (it's not slight increase).

L508: This increase is even more important in the fast and slow reservoirs.
Please also reconsider this statement. The relative increase could be large, but absolute change is larger in soil moisture.

Figure 3:

I suggest it's better to make some discussion on the water volume change and annual river discharge (by converting annual discharge to volume unit). How large the volumetric change in each reservoir is, compared to the annual discharge? This analysis must be essential to understand why discharge seasonality changed significantluy.

L551: divergent flows which very sensitive to the orography and cannot be represented in this model
Please explain why divergent flow cannot be represented. (i.e. because only one downstream is assumed for model's river network).

L639: vegetation fraction decrease
I think vegetation fraction can decrease also due to water logging along floodplains (too much water). It seems this impact is not considered in the proposed model, so better to be mentioned as limitation.

L816: . The divergent processes are not represented in the Hydrological DEM and, therefore, are not implemented in ORCHIDEE.
Divergent flow is represented in MGB-IPH and CaMa-Flood by analyzing high-resolution topography data (Pontes et al. 2017; Yamazaki et al 2014). Given that representation is possible, I think it's better to mention about the possibility.

L860: IMaps
What is IMaps? Please explain.

---

## Author Comment (AC1)

**Response to Reviewer 1**

We thank the reviewer for his/her time dedicated to this manuscript. We found the comments highly valuable to improve the quality of our manuscript.

Please see our detailed replies to each comment in blue. Text in bold is text that is copied from the new manuscript. Text in bold and highlighted in yellow is new text added as a result of the review.

This manuscript presents a new scheme for floodplains, adapted to a high spatial resolution river routing in Orchidee. The mechanism is described, and tests are performed, using two atmospheric forcing over the Pantanal wetland, between 1990 and 2013. The scheme is evaluated with river discharge in situ measurements, as well as with GRACE data and satellite-derived surface water extent. The impact of the new scheme is tested, on the soil moisture, on the surface temperature, and on the vegetation density, and on the evapotranspiration. Before being publishable, the paper has to undergo a major revision.

**Major comments**

1) How sensitive is the scheme to the dataset (here GLWD) used as a maximum mask for the inundation? A test should be performed to assess its effect, as this dataset is certainly valuable, but not perfect. There is a comment about the use of GLWD at lines 334 and following, but it is not said how the relevance of the dataset is tested (and possibly modified).

The scheme is highly sensitive to the dataset used to define the floodplains. The correct description of the flooded area is therefore essential.

To our knowledge, there are no similar global datasets differentiating the different types of wetlands. It is important to distinguish between floodplains and other type of wetlands with different hydrological dynamics.
GLWD (Lehner and Doll, 2004) characterizes all the Pantanal as potential floodplains. Therefore, we consider that the description for the Pantanal is fine and that it seems that there is no potential source of conflict with other wetland types. This is not the case of other large wetlands, which are partially floodplains (cf. answer to comment number 7).

We added the following comment in the text:
**There is a large uncertainty in the description of wetlands due to the difficulty to perfectly evaluate the flooded areas from satellite products, and there are also large uncertainties concerning the categorization. Despite this uncertainty, GLWD is combining different types of products to obtain this categorization. The review of other wetland descriptions in Hu et al. (2017) doesn't seem to show a product that would be preferable to GLWD. In this study, the GLWD dataset has not been modified, but the categories in the GLWD dataset**

**related to floodplains may be changed further in other studies to adjust the floodplains mask.**

2) Figure 2 shows an evaluation of the mean annual cycle for the discharge and the models. It would be interesting to test the inter-annual variations (directly plotting the long time series or better by calculating some de-seasonalized anomalies). Is the model able to capture these changes from a year to the next? Same question for the water masses. Is the model able to capture the inter-annual variations observed by Grace?

Thank you for this comment, this is another important aspect that can be assessed. You will find below figures performing this assessment. Figure I shows the time series of the average annual discharge at Porto Murtinho. It principally highlights the difference in terms of mean discharge over the period already plotted in Figure 2 from the paper.

Figure II shows that variations in the FP simulations is less noisy than NOFP simulations which have more important variations compared to the annual cycle, i.e. FP has a more stable annual cycle. Also, FP de-seasonalized monthly discharge time series is closer to the observations than NOFP.

We decided to include these figures in Annex, and we added the following comment in the text:

**The interannual variability has also been assessed and is shown in Figure I. The FP simulations with floodplains have higher correlations with observations compared to the NOFP simulations concerning the interannual variability of the mean annual discharge. However, these correlations are only significant for WFDEI_GPCC simulations. Also, this correlation is much higher in WFDEI_GPCC_FP (correlation of 0.71) compared to AmSud_GPCC_FP (correlation of 0.17).**
**Figures II shows the de-seasonalized time series of the monthly discharge at Porto Murtinho. We can observe that the FP simulations are less noisy and much closer to the observations compared to the NOFP simulations.**

[Figure]

*Figure I: Time series of the annual average of the discharge at Porto Murtinho between 1990 and 2013.*

[Figure]

*Figure II: Time series of the monthly discharge at Porto Murtinho removing the annual cycle between 1990 and 2013 for (a) the simulations without floodplains and (b) with floodplains.*

3) Between the two forcing datasets, the differences in terms of water masses are particularly striking (Figure 3), and as large as the difference between the cases with and without floodplains for the WFDEI case (see for soil moisture or for the slow reservoir for instance). That casts some doubts on the validity of the model / forcing combination. Can you comment?

I agree with your comment that the differences in terms of water mass are quite large, with differences between WFDEI_GPCC_FP and WFDEI_GPCC_NOFP as large as differences between WFDEI_GPCC_FP and AmSud_GPCC_FP.

Two elements that can play an important role in Land Surface Models and can explain these differences.
First, the higher resolution in AmSud_GPCC is playing an important role as, due to the absence of groundwater horizontal transport scheme, the water remains along the largest river where it has the possibility to infiltrate into the soil moisture of the flooded area while in WFDEI_GPCC it can infiltrate over a much larger area.
Secondly, although they have similar precipitation, AmSud_GPCC atmospheric forcing has dryer atmospheric conditions which lead to a more important evapotranspiration in AmSud_GPCC_FP compared to AmSud_GPCC_NOFP. This can explain the fact that the soil moisture content does not increase so much between AmSud_GPCC_FP and AmSud_GPCC_NOFP.

We also want to add that AmSud_GPCC has been used in this paper because we coupled the floodplains scheme with the regional model RegIPSL, that was used to generate the AmSud_GPCC forcing over the same grid, in another study (in writing). Despite the differences with WFDEI_GPCC, this was a way to validate and evaluate the floodplains scheme over this grid.

4) Comparisons of the surface water extent are presented for different satellite-derived surface water. We need a few sentences for each dataset, to know how they have been derived and assess their possible limitations. Otherwise, there is no interest to compare to multiple products. For instance, the sensitivity of the different products to open water / vegetated water should be discussed.

Thank you for your comment. We agree that it is essential to provide some limitations to justify the use of different products. We added the following paragraph:

We use different types of satellite products to have a complete view on the flooded area. Two products have been especially constructed over the Pantanal: Hamilton (2002) and Padovani (2010) so they may be more appropriate due to the specificity of the Pantanal floodplains, however they have some limitations: Hamilton (2002) is based on a relationship between flooded area and river height established during a short and wet period and, therefore, this relationship may differ under different climatic conditions. It is also only available up to 2000. Concerning Padovani (2010) and Schrapffer et al. (2023), the limitation is the infrequent revisit of satellite (data every 6 days) and missing images due to the use of optical satellite imagery. Padovani2010 is interpolated which helps us to have an overview of the full time series of flooded areas while Schrapffer et al. (2023) gives us precise estimates for punctual satellite without any interpolations and is available up to 2013 while Padovani (2010) is only available up to 2010. Therefore, both datasets are complementary. GIEMS-2 is a global dataset and a reference in the scientific literature in

**terms of satellite estimate of the flooded area and, it has not been specifically validated over the Pantanal, but we thought it was crucial to include it here.**

5) Some mechanisms are mentioned that cannot be considered by this river flooding scheme (l. 550). Add a paragraph in the model description to mention them (section 2)?

Thank you for your comment. We provided an overview of the mechanisms not considered by the river flooding scheme in the description of the model (Section 2):

**The floodplains scheme does not include divergent flows, neither groundwater lateral flow. Also, it does not include the reduction of the vegetation due to water logging along floodplains.**

6) For the soil moisture estimates, would it be possible to add some SMOS or SMAP retrieval? For the vegetation, any tests with NDVI or other proxy for the vegetation, in terms of seasonality and inter-annuality?

Thank you for this suggestion, satellite estimates of soil moisture face large uncertainties over South America and as the formulas they rely on may not be adapted for open water surfaces / flooded vegetation such as seen by Di Vittorio et al. (2021) in the Sudd wetland. This is why we preferred to use GRACE data to assess water masses.

Concerning vegetation, we thought your suggestion was interesting, so we tried to assess it using LAI which is the main variable driving the vegetation in ORCHIDEE, it is shown in Figure III. We use the NDVI from the GIMMS dataset generated from NOAA's AVHRR and available in Google Earth Engine because it was available from 1990.

Despite the fact that we can observe well the annual cycle in both NDVI, FP and NOFP simulation, this may not help to validate the improvement of the vegetation. Also, the interannual variation of the vegetation cannot be observed in the NDVI, since it saturates for the dense canopy of the Pantanal.

[Figure]

Figure III: Comparison of the NDVI time serie from the GIMMS dataset and generated from NOAA's AVHRR with (a) AmSud_GPCC_FP and AmSud_GPCC_NOFP and also with (b) WFDEI_GPCC_FP and WFDEI_GPCC_NOFP.

7) Applying the scheme to another region and evaluating it would certainly strengthen the paper. It is rather frustrating to have global models and datasets only applied to one specific case. At least another basin that is in the same type of environment (the Orinoco?) and for one common forcing?

Thank you for your comment. Your suggestion is totally relevant. However, the flooding process in other large wetlands in South America are not always mainly driven by overflow from large rivers, as it is the case for the Pantanal. Some other type of wetlands can exist and have major influence over the flooded area, such as the swamps and flooded forest over in the Llanos de Moxos, in the Bananal and in the surrounding of the Amazon River (cf. GLWD). Also, from GLWD, in the Llanos del Orinoco, there is a region in which the flood mechanism is driven by overflow from large rivers (floodplains) but there is also an

important area in which flood mechanism is related to swamps and flooded forest processes in the South / North and East. Another difficulty is that there are not always hydrological stations which help to assess the impact of the activation of the floodplains scheme on the basin hydrological cycle.

[Figure]

Figure IV: Description of the Lake and Wetlands over (c) the Llanos de Moxos, (d) the Llanos del Orinoco, (e) the Pantanal and (f) the Niger Inner Delta floodplains from the Global Lakes and Wetlands Database (GLWD, Lehner and Döll, 2004). The location c-f are shown in (a) for the South American regions and (b) for the African regions.

However, we follow your advice and performed the analysis over the Orinoco floodplains. There is an hydrological station at the outflow of the Llano del Orinoco but there were no data available during the period of the simulations. Therefore, Figure V shows the impact of the floodplains scheme without showing the observations.

The activation of the floodplains scheme has an impact on the discharge as the annual peak of the discharge is delayed by almost one month and only a small fraction of the flooded area is represented in the output of the model. Although the correlation seems relatively high, the flooded area is importantly underestimated. This may be related to the fact that there are also important mechanisms of swamp forest / flooded forest (see Figure IV.d) and, therefore, the horizontal transfer of soil moisture and resurgence of water will be important to represent well the hydrology of the Llanos del Orinoco. However, these mechanisms are not represented in the ORCHIDEE model. As seen with the Pantanal, their absence is even more important at higher resolution and this is what we can observe through the lower flooded area in the AmSud_GPCC_FP simulation compared to the WFDEI_GPCC_FP simulation.

These Figures have been added in Annex, and we added the following comment in the text:

It is difficult to evaluate the floodplains scheme on other South American floodplains because the flooding process in other large wetlands in South America are not always mainly driven by overflow from large rivers, as it is the case for the Pantanal. Some other type of wetlands can exist and have major influence over the flooded area, such as the swamps and flooded forest over in the Llanos de Moxos, in the Bananal and in the surrounding of the Amazon River (cf. Figure IV). Another difficulty is that there are not always observations available to assess the impact of the activation of the floodplains scheme on the basin hydrological cycle (absence of hydrological stations or stations without data).

Nevertheless, an analysis has also been performed over the Llanos del Orinoco despite the absence of observation at the station at the outflow of the floodplains using both simulations between 1990 and 2013 (cf. Figure V and VI). This flood mechanism is driven by overflow from large rivers (floodplains) but there is also an important area in which flood mechanism is related to swamps and flooded forest processes in the South / North and East (cf. Figure IV). The discharge at the outflow of the Llanos del Orinoco is delayed by one month and the flooded area is underestimated due to the absence of integration of swamps and flooded forest. We can also observe the absence of coastal floodplains which are related to other floods mechanisms.

As shown from Figure IV, the Inner Niger Delta is a region adapted to evaluate the floodplains scheme is the Inner Niger Delta which is also mainly composed by "Freshwater Marsh, Floodplain" category in GLWD.

[Figure]

**Figure V: Annual cycle of the simulated discharge at the Llanos del Orinoco outflow river discharge station (Musinacio station in Venezuela) by the simulations FP and NOFP for WFDEI_GPCC and AmSud_GPCC between 1990 and 2013.**

[Figure]

**Figure VI: (a) Location of the Llanos del Orinoco region and mean flooded fraction in (b) GIEMS-2, (c) WFDEI_GPCC_FP and (g) AmSud_GPCC, as well as the (d) (respectively h)**

correlation between the flooded fraction in WFDEI_GPCC_FP (resp AmSud_GPCC_FP) and GIEMS-2 and also (e) (respectively i) the Root Mean Square Error of between the flooded fraction in WFDEI_GPCC_FP (resp AmSud_GPCC_FP) and GIEMS-2 for the period 1992-2013.

**Minor comments**

High spatial resolution river routing is mentioned at many occasions, but the reviewer could not find the information about that spatial resolution. That has to be clearly mentioned right away in the paper.

Thank you for your comment, I specified that we are using a 2km resolution DEM to construct the river routing over the different grid. The main point of the concept of high resolution routing is better defined in the companion paper Polcher et al. (2023) to which we make reference. However, I added the following:

**In this case, the routing graph have been constructed using the MERIT-Hydro dataset at a 2km resolution.**

l.61: 'such as such as'

Thank you for highlighting this mistake. It has been corrected

l.196: the notations are confusing. Clarify.

Thank you for your comment, we reformulated this sentence:

**For this reason, the slow and fast reservoirs will not be mentioned further in this paper and as the stream and floodplains reservoir of an HUT i share the same topoindex ($\alpha_{i,stream} = \alpha_{i,floodplains}$), we will refer to this common topoindex by $\alpha_i$, with $\alpha_i = \alpha_{i,stream} = \alpha_{i,floodplains}$.**

l.208: 'thRough'

Thank you for highlighting this mistake. It has been corrected.

l.327: 'the routine graphS'

Thank you for highlighting this mistake. It has been corrected.

l.339: it would help to have a map of the area, with the river, its tributaries, and the location of the reference station.

Thank you for your comment, a map has been added in the Annex.

[Figure]

**Figure VII: Description of the domain used for both simulations (AmSud_GPCC and WFDEI_GPCC) as well as the description of the Upper Paraguay River Basin region with delimitation of the Pantanal. The different rivers, regions and hydrological stations mentioned in the present articles are also described**

l.440: 'Depending on the period simulated, the SIMULATED flooded area simulated was…'

Thank you for highlighting this imprecision. It has been corrected.

Table 2: indicate the meaning of the *. It is done in Table 3, but not here.

Thank you for pointing out this omission.

l.564-565: Surfaces of point 2) are not seen by GIEMS-2. Are they seen by the mNDWI estimates?

These regions are detected by mNDWI however it is may not appear well in GIEMS-2 due to the resolution as the scale of these flooded is much smaller than the other flooded area of the Pantanal.

Figure 5: Add some comparisons with the other satellite-derived estimates. Especially the one the authors are themselves deriving.

We understand your comment. The comparison that we can make would be limited to the satellite derived estimate we derived. However, this represents a technical issue due to the much higher resolution of the satellite estimate flood map (30m resolution) compared to the output of the simulation (20km and 50km). Moreover, the interest of this figure lies in the illustration of the spatial analysis of correlation and Root Mean Square Error. However, the satellite-derived estimate haven't a regular temporal timestep and is more available during specific seasons (less cloudy season), this can potentially introduce bias into the evaluation. For this reason we only focused on GIEMS-2 which have a resolution close to the resolution of the simulations allowing to interpolate it.

l.631: 'relativeS'

Thank you for highlighting this mistake. It has been corrected.

l.798: 'assess flooded area…principally in areas covered by floods'????

Thank you for highlighting this mistake. It has been corrected. We meant "covered by vegetation".

l.806: All the satellite-products do not only consider the open-water surfaces. In this work, the model is expected to be evaluated for wetlands. Most wetlands are vegetated surface water. If the satellite-products you use are only sensitive to open-water, it seems that the paper is missing its goal. Clarify.

Just to be more precise, this model is expected to be evaluated on floodplains because the floodplains scheme is not able to represent the processes occurring in other types of wetlands (such as swamps, for example).

Concerning the vegetation, it depends on the type of vegetation, as satellite estimates of flooded areas such as Padovani (2010) have succeeded in identifying the flooded areas over the Pantanal. The issue with these satellite products is not there, the issue is that it can confuse areas with saturated soil but no flood with flooded areas because they detect an important presence of water over the soil. Therefore, these satellite products are not only sensitive to open water, but they are sensitive to saturated soil which are not flooded.

**REFERENCES**

Di Vittorio, C. A., & Georgakakos, A. P. (2021). Hydrologic modeling of the sudd wetland using satellite-based data. Journal of Hydrology: Regional Studies, 37, 100922.

Hamilton, S. K.: HYDROLOGICAL CONTROLS OF ECOLOGICAL STRUCTURE AND FUNCTION IN THE PANTANAL WETLAND (BRAZIL), IAHS Special Publication. The Ecohydrology of South American Rivers and Wetlands., 2002.

Hu, S., Niu, Z., & Chen, Y. (2017). Global wetland datasets: a review. Wetlands, 37, 807-817.

Lehner, B., & Döll, P. (2004). Development and validation of a global database of lakes, reservoirs and wetlands. Journal of hydrology, 296(1-4), 1-22.

Padovani: Dinâmica Espaço-Temporal das Inundações do Pantanal., Ph.D. thesis, Piracicaba: Escola Superior de Agricultura Luiz de Queiroz, Centro de Energia Nuclear na Agricultura, Universidade de São Paulo, http://www.teses.usp.br/teses/disponiveis/91/91131/tde-14022011-170515/pt-br.php, 2010

Schrapffer, A., María Cappelletti, L., and Sörensson, A.: ESTIMATION OF THE FLOODED AREA OVER THE PANTANAL, A SOUTH AMERICAN FLOODPLAIN, USING MODIS DATA., Meteorologica, 48, 2023

---

## Author Comment (AC2)

**Response to Reviewer 2**

We thank the reviewer for his/her time dedicated to this manuscript. We found the comments highly valuable to improve the quality of our manuscript.

Please see our detailed replies to each comment in blue. Text in bold is text that is copied from the new manuscript. Text in bold and highlighted in yellow is new text added as a result of the review.

**General Comments**

This manuscript describes the new floodplain scheme implemented in ORCHIDEE model, evaluates the validity of the new scheme, and analyzes its impact on other land surface variables. Even though it's still a case study simulation over Pantanal, I feel the paper very carefully analyzed how floodplain is important for land surface modeling.

The modeling strategy seems to be a bit complicated, while I feel the complexity is necessary given that the floodplain inundation itself is a complex physical process. I suggest the authors to provide more kind explanations about floodplain parameterization scheme, for example by using schematic figures, to help readers to understand how the proposed floodplain scheme works. However, the manuscript is overall well written, while minor revision is needed before acceptance.

**Major concerns**

[1] I feel the manuscript is too long. It might be unavoidable as a model description paper, but readability might increase if not-so-important parts are moved to supplements.

We agree with your comment, there has been an important effort of reducing the text and of moving figures into the supplement section before submitting the initial version. We kept this issue in mind when integrated the changes related to the reviewers' comments.

[2] So many variables/symbols are used to parameterize the proposed floodplain scheme, and I feel difficulty following the explanations and equations. I suggest creating one schematic figure which represents the parameterization concept of floodplain scheme (with explicit description of which symbols correspond to which variables). Visual explanation must help readers to understand about the new floodplain scheme.

Your comment is totally relevant, most of the variables were present in Figure 1, however the name of fluxes and reservoirs in this figure did not correspond anymore with the text. We decided to update this figure harmonizing the name of the variables with the names used in the equation and adding the variables that were not present, such as Evapotranspiration and Precipitation over floodplains and infiltration from floodplains.

[Figure]

**Figure 1: Scheme summarizing the movement between the different reservoirs for a HTU which has floodplains and its upstream HTUs if (a) the upstream HTU has floodplains or if (b) the upstream HTU doesn't have floodplains and (c) the fluxes between the HTU, the atmosphere and the soil moisture.**

**Specific comments:**

L193: whether the floodplains are activated or not.

This should be "regardless of whether ..."

Thank you, the text has been corrected.

In addition, please explain what slow and fast reservoir represent. It is explained in the results section that they represent aquifer and shallow groundwater, but this should be stated here. Otherwise, readers cannot know why they have limited relationships to floodplain scheme.

We agree with your comment, we have changed the description to add these details:

Each HTU contains four water reservoirs used by the river routing scheme to represent processes with different time constants: the stream reservoir for the river flow processes, the fast reservoir receiving the surface runoff, the slow reservoir which receives the deep

**drainage and the floodplain reservoir. The fast and slow reservoirs can be viewed respectively as a conceptual representation of the rapid shallow aquifer and the slower deeper one.**

L235: The floodplains scheme allows a specific HTU to "overflow" the content of its floodplains reservoir into connected upstream HTUs with floodplains.

This is a very interesting scheme. I wonder what is the impact of this overflow scheme on simulated water and energy budgets. If space allows, please include some analysis.

The energy and water balance are performed at the level of the grid cell, therefore it is difficult to assess the impact of the overflow. The impact that can be distinguished is for the overflow, transporting water from a HTU in a grid cell to another HTU in another grid cell.

The best option to perform this analysis would have been to perform an additional simulation without overflow to compare it. This can be an interesting experiment in future studies with the floodplains scheme. However, we haven't performed such an experiment and the content of the paper is already very dense.

We also have thought to track the fluxes of overflow, but this was technically impossible because this would have represented a very large amount of data because this would have been saved in the HTU grid (even the discharge is not saved at the HTU level we only save it for a limited number of stations).

L284 2.4.1: Case $S_{f,I} < S_{fmax,i}$

I recommend you to explain the case in plain language in the section title, not by the equation.

Thank you for this comment, we changed the title to more explicit version of them:
**Cases of not fully flooded floodplains**

**Cases of fully flooded floodplains**

L285: height of the floodplain

This term is ambiguous. Do you mean "water surface elevation of the floodplain"?

You are right, we changed the formulation.

L331: in order to define a mask of potentially flooded areas based on the following categories:
Could you please explain in which case this floodplain mask is required, and what is the impact of using this floodplain mask?

The floodplains mask is required when there is a process of flooding, mainly driven by overflow of a river.

The objective of the floodplains is to identify the regions which are susceptible to flood due to the presence of a river. Among the existing categories, the one which better fit is the "freshwater marsh, floodplain". We also decided to include the reservoir to capture the existing flooded existing along the Paraná river and which flood is driven by the river.

L355: before using the scheme over another region to evaluate if this parameterization is the more appropriate.

In many parts of the world, there is no observation data for calibration. If possible, it's better to perform some sensitivity tests of parameters (confirm results are not so sensitive to parameters, or specify which parameter has larger impact).

We agree with your comment, we reformulated the subsection about calibration to clarify the sensitivity of the different parameters.

**The different parameters of the floodplains scheme have been calibrated based on the simulated discharge at the *Porto Murtinho* station, which is the reference station at the outflow of the Pantanal (Brazil, lat: 21.7°S, lon: 57.9°W) between 1991 and 1996 in comparison to the observations considering: (1) the variation of the discharge through its correlation with the observations and (2) the mean value and variability of the discharge. The choice of the 6 years calibration period was due to a limited number of available years from the simulations (24 years). Therefore, the model has been calibrated over this reduced period common to both forcing in order that the results analyzed after are not influenced by an overfitting effect. Considering that our model have a reduced number of physical variables, we consider it is not necessary to assess it on large periods as we made the assumption that these parameters are relatively independent of the hydrological cycle variability. However, we agree that performing the calibration over a larger period of time could have been preferable, but we faced 2 limitations for this point: 1) the period of the simulations (AmSud was only available from 1990 to 2019) and 2) a technical limit due to the resources (time and computational resources) needed to run the simulations.**

**The parameter with the largest influence on the variability of the discharge is $\tau_f$, the time constant of the floodplains reservoir. This parameter has an important impact on the annual cycle of the discharge at Porto Murtinho station. The $[\alpha_{stream}, \alpha_{fast}]$ interval is considered as a valid interval for $\tau_f$. This interval has been discretized to select different possible values for $\tau_f$.**
**It has been assessed along with $R_{limit}$ which is the second parameter with the largest influence on the discharge. For $R_{limit}$, we discretized the [0,1] interval to obtain possible values.**
**In a first step, these two parameters have been calibrated together, we performed a grid-search evaluation, which means that we evaluated all the existing combination of possible discretized values over the intervals for $\tau_f$ and $\tau_f$ to select the combination with the best performance to represent the observed discharge.**

**In a second step, we assessed the parameters related to the overflow, which have a limited impact on the discharge $OF$ and $OF_{repeat}$. These parameters slightly**

**influence the temporality of the discharge. In this case, we also assessed these two parameters using a grid-search evaluation considering a discretization of the following intervals: [0.5 day, 2 days] for $OF$ and [1 repetition, 5 repetitions] for $OF_{repeat}$.**

**Finally, the last parameter to calibrate is the infiltration constant ($C$) which determines the loss to soil moisture and, thus, potentially to evaporation. This parameter with a very reduced impact on the discharge and only reduce / increase the level of the discharge at the outflow of the region. We discretized the [0,1] interval to assess it.**

L360: Methodology of Validation and Analysis

Please also provide some description of the simulation domain. Probably, a figure showing the simulation domain (with location of the gauges) is better to be provided.

Thank you for your suggestion, we agree that this should be included, the following Figure I has been added in Annex.

[Figure]

1. Main Paraguay River
2. São Lourenço
3. Cuiabá
4. Taquari river
5. Taquari megafan
6. Nhecolândia

**Figure I: Description of the domain used for both simulations (AmSud_GPCC and WFDEI_GPCC) as well as the description of the Upper Paraguay River Basin region with delimitation of the Pantanal. The different rivers, regions and hydrological stations mentioned in the present articles are also described**

L419: forced with ERA5 re-analysis data.

I assume this is regional atmospheric simulation, and in that case ERA5 must be "boundary condition" rather than "forcing".

You are right, this has been clarified.

Figure 2:
Could you please analyze the mechanics of river discharge delay? E.g. where water stays before reaching to the river gauge? Did they stay in floodplain as surface water? Or did they stay in soil by infiltration? Given that the difference between FP and NOFP simulation is large, it's better to provide detailed analysis on the mechanism which cause the difference.

Thank you for your comment, we specified the following in the analysis:

The main mechanism behind the river discharge delay is that the water is delayed in the floodplain reservoir. Another part of the delay is also related to the infiltration of the water in the floodplains into the soil, which face a larger delay. Then, the evapotranspiration also plays an important role as it will reduce the mean annual river discharge.

L507: soil moisture and in the stream reservoir increases slightly

Considering the magnitude of change, compared to other storage variables, I feel the soil moisture was "significantly" increased by floodplain scheme (it's not slight increase).

Thank you for your comment, we agree that it is not the right term as the increase is strong compared to volume of water in other reservoirs. This has been corrected.

L508: This increase is even more important in the fast and slow reservoirs.

Please also reconsider this statement. The relative increase could be large, but absolute change is larger in soil moisture.

We think that your comment is relevant, we changed the text accordingly.

Figure 3:

I suggest it's better to make some discussion on the water volume change and annual river discharge (by converting annual discharge to volume unit). How large the volumetric change in each reservoir is, compared to the annual discharge? This analysis must be essential to understand why discharge seasonality changed significantly.

Thank you for your comment, we added the discharge in Figure 3 as a reference.

L551: divergent flows which very sensitive to the orography and cannot be represented in this model

Please explain why divergent flow cannot be represented. (i.e. because only one downstream is assumed for the model's river network).

Thank you for your comment, we added this precision.

**[...] which is an area of divergent flows which very sensitive to the orography and cannot be represented in this model (Louzada et al., 2020; Assine, 2005)** because the model's river network is convergent and only assumes a downstream.

L639: vegetation fraction decrease

I think vegetation fraction can decrease also due to water logging along floodplains (too much water). It seems this impact is not considered in the proposed model, so better to be mentioned as a limitation.

You are right, vegetation can also decrease due to water logging along floodplains, however this is not included in the model. We provided an overview of the mechanisms not considered by the river flooding scheme in the description of the model (Section 2):

**The floodplains scheme does not include divergent flows, neither groundwater lateral flow. Also, it does not include the reduction of the vegetation due to water logging along floodplains.**

L816: . The divergent processes are not represented in the Hydrological DEM and, therefore, are not implemented in ORCHIDEE.

Divergent flow is represented in MGB-IPH and CaMa-Flood by analyzing high-resolution topography data (Pontes et al. 2017; Yamazaki et al 2014). Given that representation is possible, I think it's better to mention about the possibility.

You are right, there are some models integrating this possibility, we corrected this part by adding that there are divergent models and quoted some examples.

**The divergent processes are not represented in the Hydrological DEM and, therefore, are not implemented in ORCHIDEE. However, some models such as MGB-IPH and CaMa-Flood represent this divergent process by analyzing high resolution topography data (Pontes et al. 2017; Yamazaki et al 2014).**

L860: IMaps

What is IMaps? Please explain.

Thank you for highlighting this mistakes, it is replaced by "**Spatial description of wetlands**".

**Bibliography**

Pontes, P. R. M., Fan, F. M., Fleischmann, A. S., de Paiva, R. C. D., Buarque, D. C., Siqueira, V. A., Jardim, P. F., Sorribas, M. V., and Collischonn, W.: MGB-IPH model for hydrological and hydraulic simulation of large floodplain river systems coupled with open source GIS, Environmental Modelling and Software, 94, 1–20, https://doi.org/10.1016/j.envsoft.2017.03.029, 2017

Yamazaki, D., Sato, T., Kanae, S., Hirabayashi, Y., & Bates, P. D. (2014). Regional flood dynamics in a bifurcating mega delta simulated in a global river model. Geophysical Research Letters, 41(9), 3127-3135.

---

## Author Comment (AC3)

**Response to Reviewer 3**

We thank the reviewer for his/her time dedicated to this manuscript. We found the comments highly valuable to improve the quality of our manuscript.

Please see our detailed replies to each comment in blue. Text in bold is text that is copied from the new manuscript. Text in bold and highlighted in yellow is new text added as a result of the review.

This paper describes a new floodplain scheme developed within the framework of the land surface modeling platform ORCHIDEE. The main applications of this new model development are intended to be used at the regional-to-global scale in so-called "offline mode" (decoupled from a regional climate, RCM, or global-scale earth system/climate model, GCM) or coupled to an atmospheric model, thus the level of complexity, process representation and input data are adapted for such applications. As noted by the authors, RCM and GCM spatial resolutions are constantly increasing, thus there is a need to adapt the hydrological parameterizations in such models accordingly. Rather than using a classic grid structure (as many GCMs currently use) dictated by the atmospheric model, the current scheme is based on the Hydrological Transfer Unit (HTU) concept. The implementation of this scheme benefits from numerous relatively high spatial resolution topographical and geomorphological off-the-shelf datasets now available to hydrologists. This paper describes the methodology and mathematical underpinnings of this new floodplain scheme and how it interacts with other components of ORCHIDEE (such as evaporation, river flow, runoff, etc.). The scheme is next used to simulate the floodplains along with the other main components of the surface hydrological cycle over a recent multi-year period over the Pantanal basin in South America, which contains one of the world's largest floodplains thus making it a very pertinent case study. The model is evaluated at two spatial scales, one representing the approximate scale still used by many GCMs (i.e. 0.5 o) and another representing a scale comparable to RCMs and what more and more GCMs are (or plan to) move to in upcoming years (~25 km). As boundary conditions, so-called atmospheric forcing must be prescribed in offline mode but there are many such products and there are considerable differences among them, especially at different spatial scales as herein. The authors have addressed these uncertainties by using a very standard analysis product as forcing at the more coarse resolution, along with a forcing which has been developed specifically for this region at a higher spatial resolution. The model simulations, notably the floodplain outputs, are evaluated using several standard satellite-based products along with in-situ discharge measurements. Convincing statistical results are used to summarize the performance of the model using the new parameterization for the two input forcing compared to the baseline model (without the new floodplain scheme). A discussion of errors (in terms of the model input, output and the evaluation data), limitations, and gained insights are presented. I find the organization of the paper to be quite good, it is well written: the overall presentation is clear, the results are presented in a very pragmatic manner and future perspectives are discussed. I recommend publication after only some minor revisions as this paper is an important contribution to the rapidly developing region-to-global scale hydrological modeling field, notably improved terrestrial water cycle simulations in RCMs and GCMs.

**General Comments:**

1. Lines 339-358: In my opinion, the only part of this paper which needs some improvement is Calibration of the Parameters. There are no graphics (for example, showing the discharge performance at the calibration station) and only limited statistics (Table 1.).

Lines 347-349 mention that The best combination of parameters has been established through a grid search method which consists in evaluating the different combinations of parameters within their respective interval of definition. I find this a bit vague and it seems to gloss over a very important part of any new parameterization: parameter calibration/estimation/determination. I feel the authors should just give a slightly more detailed description of how exactly the parameters were calibrated. There is some limited information, but more details would be appreciated. Also, plots of discharge before and after calibration would be informative. Also, 1991-1996 was the calibration period: why these 6 years? Is the natural variability adequately represented over these years? And so on. Again, just a few more details on the methods and results. Parameter sensitivity analysis is a critical part of any new model development and a bit more information would be very informative to readers.

We appreciate your comment, and as per your suggestion, we have completely rewritten this section.

We didn't include figures of the before and after calibration process because there is no "initial state" of the parameters, we directly compared the outputs for a range of values which was estimated as physically reasonable. This is why we focused our analysis on the comparison of the discharge with and without the floodplains scheme activated, this would be equivalent to a floodplain time constant equal to the stream reservoir time constant, a very large OF parameter, a flood fraction always equal to zero and a C parameter equal to 1.

The new version of this subsection about the calibration emphasize the role of each parameter, how they affect the simulated discharge and the model in general and the relative sensibility of the simulated discharge to each parameter and then described.

**The different parameters of the floodplains scheme have been calibrated based on the simulated discharge at the *Porto Murtinho* station, which is the reference station at the outflow of the Pantanal (Brazil, lat: 21.7°S, lon: 57.9°W) between 1991 and 1996 in comparison to the observations considering: (1) the variation of the discharge through its correlation with the observations and (2) the mean value and variability of the discharge. The choice of the 6 years calibration period was due to a limited number of available years from the simulations (24 years). Therefore, the model has been calibrated over this reduced period common to both forcing in order that the results analyzed after are not influenced by an overfitting effect. Considering that our model have a reduced number of physical variables, we consider it is not necessary to assess it on large periods as we made the assumption that these parameters are relatively independent of the hydrological cycle variability. However, we agree that performing the calibration over a larger period of time could have been preferable, but**

we faced 2 limitations for this point: 1) the period of the simulations (AmSud was only available from 1990 to 2019) and 2) a technical limit due to the resources (time and computational resources) needed to run the simulations.

The parameter with the largest influence on the variability of the discharge is $\tau_f$, the time constant of the floodplains reservoir. This parameter has an important impact on the annual cycle of the discharge at Porto Murtinho station. The $[\alpha_{stream}, \alpha_{fast}]$ interval is considered as a valid interval for $\tau_f$. This interval has been discretized to select different possible values for $\tau_f$.
It has been assessed along with $R_{limit}$ which is the second parameter with the largest influence on the discharge. For $R_{limit}$, we discretized the [0,1] interval to obtain possible values.
In a first step, these two parameters have been calibrated together, we performed a grid-search evaluation, which means that we evaluated all the existing combination of possible discretized values over the intervals for $\tau_f$ and $\tau_f$ to select the combination with the best performance to represent the observed discharge.

In a second step, we assessed the parameters related to the overflow, which have a limited impact on the discharge $OF$ and $OF_{repeat}$. These parameters slightly influence the temporality of the discharge. In this case, we also assessed these two parameters using a grid-search evaluation considering a discretization of the following intervals: [0.5 day, 2 days] for $OF$ and [1 repetition, 5 repetitions] for $OF_{repeat}$.

Finally, the last parameter to calibrate is the infiltration constant ($C$) which determines the loss to soil moisture and, thus, potentially to evaporation. This parameter with a very reduced impact on the discharge and only reduce / increase the level of the discharge at the outflow of the region. We discretized the [0,1] interval to assess it.

2. The quality of the English is good, however there are a certain number of very small errors, notably the use or lack thereof of "a" or "s" at the end of some words, e.g. Line 48: a South American tropical floodplains. There are just a few small errors like this on nearly every page, so they do not detract from the reading or result in a lack of understanding. But I'd recommend a quick filtering to catch them.

Thank you for your comment, we performed a complete review to identify and correct these issues.

**More specific:**

Line 121: I suggest changing ruling to governing

Thank you, the text has been corrected.

Line 130: Referring to the text: HTU only flows into a single HTU and is acyclic as water cannot return to the original HTU: I assume that backwater effects can be neglected at the spatial resolutions you are modeling here?

Exactly, backwater effects are neglected because they are not relevant at this resolution. However, they can have an impact over larger river such as at the confluence of Paraná and Paraguay river, but this is out of our area of interest.

Eq.2 for evaporation from the floodplains: water surfaces have very low roughness lengths compared to land surfaces: typically Charnock-type parameterizations are used for water bodies. I assume that floodplains are generally fairly smooth...should this effect (or is it?) somehow incorporated into this computation? I suspect that using such a roughness length could reduce the evaporation from floodplains (?).

Thank you for your comment.
The Charnock-type formulation for surface roughness is conceived for open oceans without any surface elements (except waves) which can generate atmospheric turbulence. In the case of the floodplain or a lake, the open water is surrounded by trees or mountains, generating turbulence over the open water. It is thus not a given that the open water of a floodplain or lake has the same effective roughness as the open ocean.
We can evaluate the use of this type of formulation over flooded areas in future works.

Line 172: I am surprised that soil water infiltration can be larger outside of floodplains than within them. Can the authors present some sort of physical arguments or an observational basis for this assumption?

We agree that this point is not clear. The k_litt parameter is the Hydraulic conductivity at saturation over the first layer of soil. We assumed that this parameter can change over the floodplains because these processes at the interaction between flood water and soil can be altered due to the presence of sediments which can decrease the infiltration rate. This is why we decided to open the possibility to calibrate this parameter. The outcome is that this parameter was identical in the higher resolution simulation and has been found lower in the WFDEI_GPCC simulation.

This was not originally clarified, and you are right that we need to be more transparent on our original assumptions. This has been clarified:

**This k_litt parameter has been established for the soil infiltration processes but not specifically for floodplains. Therefore, we assume that the infiltration can be different over the floodplains due to the presence of sediments, which may reduce the infiltration capacity. This is why a reduction factor (C) has been introduced to evaluate changes in the infiltration over flooded areas if necessary. This parameter may**

**depend on the local properties of the region considered such as the type of vegetation or the soil and the sediments which cannot be represented explicitly.**

Line 188: Referring to the text: The time constant of the floodplains (τf) is slower than the stream reservoir time constant (τstream) and faster than the fast reservoir time constant. Can the authors give some sort of physical argument or explanation for this (frictional effects of flooded riparian vegetation and non-riparian vegetation in flooded zones for example? Or some other reason? Or just a reference justifying this choice?)

The floodplains time constant is necessarily higher than the stream reservoir time constant because the floodplains reservoir represents the slow-down of the river discharge flow over the floodplains. Still, the time constant in the floodplains is related to the river flow and, therefore, should be lower than runoff processes. This difference between the stream time constant and floodplains time constant is related to frictional effects of flooded riparian vegetation and non-riparian vegetation in flooded zones due to the locally divergent flow of water sparsing.

The following sentence has been added in the article:
**The time constant of the floodplains (τf) is slower than the stream reservoir time constant (τstream) and faster than the fast reservoir time constant because the dynamic floodplains reservoir represents the slow-down of the river flow over the floodplains due to frictional effects of flooded riparian vegetation and non-riparian vegetation in flooded zones due to the locally divergent flow of water sparsing. The fast reservoir model a slower dynamic related to runoff and therefore is an upper limit for the floodplains reservoir time constant**

Eq.5: It is not quite clear to me why when Sfmax,i > 0 there is no contribution from the upstream stream reservoir to the local stream flow (it is just from the upstream floodplain...)...I am missing something here.

When a certain HTU is considered as a floodplain (Sfmax,i > 0) the water is not coming from the stream reservoir of the upstream HTUs but first flow into its floodplain reservoir. (Qf,i).

Eq.8: It seems that a term is missing on the RHS...the possible addition of overflow from the downstream reservoir?

You are right, thank you for highlighting this omission.
The possible overflow of the current floodplains HTU into the upstream one is not included there. This has been corrected.

Lines 271-273 should probably be placed after Eq.13 since "beta" doesn't seem to be mentioned until Eq.13.Lines 312-313: Referring to the text: The different values of standard deviation are bounded by lowlim_std = 0.05m and uplim_std = 20m. Why these particular values? Is the model very sensitive to this range?

Concerning the first point, we think that it is better to keep the description of beta along with the other variable used to describe the floodplains geometry in 2.4. This way, beta is also defined before using it in the equations in the following subsection in 2.4.1 (eq. 13).

Concerning the second point, it was difficult to establish a simple relationship between the distribution of the elevation within a HTU and this beta variable. We used clustering methods to analyze the different type of distribution and the corresponding beta. This is how we defined these limits and we are aware that this is a raw approximation and this is something that need to be improved in future development of the model. The simulation of the discharge is not so sensitive to these values, however this will affect the flooded area but in a small extent.

Line 465: It seems that there are only roughly 1 to 2 GRACE pixels covering your zone, likely not with a perfect overlap. Is this really sufficient? Can you say a bit more about the errors involved in this comparison to justify this for readers?

You are right and we are aware of this. However, GRACE was the best tool we had at this moment to perform this type of analysis. Although the area is large enough to justify the use of GRACE, there can be, as you say, overlapping error. It is more of a qualitative comparison than a quantitative one. Hopefully, the new generation of GRACE will allow seeing it more clearly.

We added the following specification:
**Although the area is large enough to justify the use of GRACE, there can be an error related to the overlap of pixels. Still, GRACE is the best tool available at this moment to perform this type of analysis. Also, the comparison GRACE is more of a qualitative than a quantitative one.**

Line 475: Maybe I missed it, but I assume the statistics were made using monthly model outputs and observations?

Exactly, thank you for highlighting this imprecision. This has been specified.

Fig.3 showing the multi-year monthly averages as a single annual cycle is indeed an informative way to convey the quality of the climatological performance of the scheme. But aside from the statistics, it would be good to see some graphical information on the year-to-year variability per month in the main paper: some sort of spread (standard deviation or quantiles, etc.) on these plots would be most informative. Indeed, we wish to see the climatological (average annual cycle), but it is of course the improvement or degradation in terms of model vs observed variability that is also of interest.

Thank you for this suggestion, Figure I has been added in Annex and the following comment has been added in the article:
**The interannual variability of the monthly discharge at Porto Murtinho is shown in Figure I. We can observe that the floodplains scheme reduces the variability of the discharge. Between October and April, the variability of the FP simulations is closed to the observed discharge variability. From May to September, the variability of the**

**monthly discharge is overestimated compared to the observation. This overestimation is higher in WFDEI_GPCC_FP compared to AmSud_GPCC_FP.**

[Figure]

**Figure I: observed and simulated boxplot representing the interannual variability of the average monthly discharge at Porto Murtinho between 1990 and 2013.**

Line 522: Referring to the text: AmSud_FP seems to have more runoff. This sounds a bit speculative and it seems that it would be easy to verify by comparing the modeled runoff with and without floodplains? The authors could just include some numerical values here within the text for example.

Thank you for highlighting this imprecision, the sentence was not clear.

We also quantified this aspect more precisely and calculated the range of order of the runoff in the different simulations over the Pantanal floodplains. The runoff and drainage and is higher in the simulations with floodplains activated. The runoff is 3 times higher (respectively 63 times higher) in the AmSud_GPCC_FP (respectively WFDEI_GPCC_FP) simulation compared to the AmSud_GPCC_NOFP (respectively WFDEI_GPCC_NOFP) simulation. The higher increase in WFDEI_GPCC can also be observed in the fast reservoir difference between WFDEI_GPCC_FP and WFDEI_GPCC_NOFP in Figure 3.f.

The sentence you mention has been removed and replace by the following :

**This can be explained by the increase of runoff in the FP simulations compared to the NOFP simulations (not shown). This increase is much higher in WFDEI_GPCC than in AmSud_GPCC.**

End of Section 4.2: After reading this section, I am left wondering whether it possible to give a number or show a figure of the contribution of Eflood to the total E? The total E with floodplains will almost certainly increase (when using the same prescribed forcing) over soils

which have been wetted once floodwaters retreat, so increases in E will be at least related to this, as discussed in the paper. But Eflood seems to be rather uncertain/difficult to model and observe, I wonder how much Eflood is contributing to the overall E increase. I say this because I wonder if a more surface-water adapted approach for E might be in order, especially if this flux is significant compared to the other E components.

Thank you for this comment. The different components of evapotranspiration can be found in Figure 10. As discussed with your previous comment on the Charnock-type parameterization, the limitation of this type of approach will be the important local surface heterogeneities and also the presence of vegetation over and close to the flooded area.
As observed in Figure 10.c, if the floodplains lead to a significant soil moisture increase (such as it is the case with the WFDEI_GPCC forcing) the transpiration can have a non-negligible role in the increase of the evapotranspiration during the dry season. It is as important as direct evaporation from the flooded area during this season.

Lines 615-620: If I understand correctly, rainfall can lead to greater soil water infiltration that when the same grid element is flooded. This seems a bit counter-intuitive, to me anyway. Are there observational studies which can be referenced etc. to justify this?

This is a side effect of the modeling choices. Over a slightly flooded grid point, the water in the precipitation will go partly to the floodplain reservoir and in the case the flooded area is small, a lower volume of water will infiltrate. When there are no floodplains, all this water goes directly to soil moisture.

Lines 670-674: What about the increase in net radiation over the flood waters? The typical albedo for water surfaces is generally around 0.07, far lower than vegetation or soil. Is this considered?

This is due to the lower surface temperature. The surface albedo is not yet changed by the floodplains, but it should be !

Line 676-677: typo, a phrase is repeated → depending on vegetation type and on soil types (Clay, Sand, Silt). depending on vegetation and soil types (clay, sand, silt)

Thank you for highlighting this repetition. It has been corrected.

Lines 730-731: Can the authors just provide a phrase describing the specific sub-surface component and how this could help solve the mentioned issues?

We agree that this brings values to the discussion to detail a bit more the content of this sub-surface component and clarify how this solves the mentioned issues.

**The use of a specific sub-surface component such as suggested in the framework for LSM described in Hallouin et al. (2022) can be used to solve these issues by providing a tridirectional movement of the water in the ground with a lateral movement driven by topographic and hydraulic head gradients.**

**Reference**

Hallouin, T., Ellis, R. J., Clark, D. B., Dadson, S. J., Hughes, A. G., Lawrence, B. N., ... & Polcher, J. (2022). UniFHy v0. 1.1: a community modelling framework for the terrestrial water cycle in Python. Geoscientific Model Development, 15(24), 9177-9196.

---

## Author Response (AR2)

**Response to Reviewer 1—Report #2**

Please see our detailed replies to each comment in blue. Text in bold is text that is copied from the new manuscript. Text in bold and highlighted in yellow is new text added as a result of the review.

The authors have responded to my comments thoroughly and successfully. Their effort to test the model on other basins is much appreciated.

I would only suggest that for Figure III they use two different scales (left and right of the figure) for their estimates and for the satellite-derived NDVI, to facilitate comparison of the inter-annual variations. They could also calculate time correlations between the modeled and satellite derived variables.

Thank you for this suggestion, we improved Figures III according to it, we also added the correlations.

[Figure]

Figure III: Comparison of the NDVI time serie from the GIMMS dataset and generated from NOAA's AVHRR with (a) AmSud_GPCC_FP and AmSud_GPCC_NOFP and also with (b) WFDEI_GPCC_FP and WFDEI_GPCC_NOFP over the Pantanal region. Correlations between the simulated LAI and the NDVI are shown in legend, if the significance level is higher than 95%, there is a star symbol after the correlation.

We decided to include the figure with the correlations in the Supplementary material and mentioned it in the paper as the following:

As a qualitative assessment, the average simulated LAI over the Pantanal is compared to the Global Inventory Modeling and Mapping Studies-3rd Generation V1.2 (GIMMS-3G+) data for the Normalized Difference Vegetation Index (NDVI) (Pinzon et al. 2023) in Figure III. The LAI time series have significant correlations with the NDVI time series but the NOFP simulation have a higher correlation compared to FP simulations. This seems to be caused by the delayed peak of LAI in the FP simulations.

It is also advisable to proofread the full article carefully, as a few typos can still be noticed.

Thank you for noticing this. We also performed a review of the text to correct remaining typos.

**References**

Pinzon, J.E., E.W. Pak, C.J. Tucker, U.S. Bhatt, G.V. Frost, and M.J. Macander. 2023. Global Vegetation Greenness (NDVI) from AVHRR GIMMS-3G+, 1981-2022. ORNL DAAC, Oak Ridge, Tennessee, USA. https://doi.org/10.3334/ORNLDAAC/2187